# *Stable-Transformer*:
# TOWARDS A STABLE TRANSFORMER TRAINING

## ABSTRACT

The scale of parameters in Transformers has expanded dramatically—from hundreds of millions to several trillion. A key challenge when scaling the model to trillions is the training instability. Although many practical tricks, such as learning rate warmup, query-key normalization and better weight initialization, have been introduced to mitigate the training instability, a rigorous mathematical understanding of why such instabilities happen and why the above-mentioned tricks work well is still unclear. In this paper, we give a theoretical analysis of the initialization, normalization and attention mechanism in Transformers, and present a set of stabilized designs of the initialization, normalization and attention mechanism, which are thus termed as *StableInit*, *StableNorm* and *StableAtten*, individually. In experiments, we demonstrate that each of our stabilized designs, *i.e.*, *StableInit*, *StableNorm* and *StableAtten*, exhibits better stability. Furthermore, by putting the stabilized designs together, we propose a stabilized Transformer, termed *Stable-Transformer*, and show in experiments on large model (1B parameters) and deep model (200 layers) that our proposed *Stable-Transformer* achieves a more stable training process.

> *"My work always tried to unite the truth with the beautiful, but when I had to choose one or the other, I usually chose the beautiful."*
>
> *— Hermann Weyl*

## 1 INTRODUCTION

The scale of parameters in Transformers (Vaswani et al., 2017; Radford et al., 2018; 2019; Brown et al., 2020; Touvron et al., 2023; Chowdhery et al., 2023) has expanded dramatically—from hundreds of millions to several trillion—parallel to significant advancements of hardware capabilities in the field of deep learning (Goodfellow et al., 2016; LeCun et al., 2015; Bengio et al., 2021). This exponential growth in model size has been facilitated by equally significant strides in computational power, enabling deeper and more complex network architectures. As these models have grown, they have set new benchmarks across a myriad of tasks in various fields such as natural language processing (Dubey et al., 2024; Achiam et al., 2023), computer vision (Ravi et al., 2024), and generation (Peebles & Xie, 2023).

Despite of these significant achievements, training larger models still suffers from an instability issue, which is often characterized as the difficulties in convergence, the sensitivity to initial conditions, and the necessity to finely tuned optimization strategies. Since that the instability in training process encumbers the deployment and real-world applicability of the sophisticated models, it is crucial to have a mathematical understanding of why such instability happens and it is urgent to invent stabilized design of the architecture or training strategies.

To gain a deeper understanding of the instability in training Transformers, it is essential to investigate the training dynamics of Transformers. To date, there are various studies devoted to the training process of Transformers from different perspectives, including normalization (Wang et al., 2019; Xiong et al., 2020; Liu et al., 2020; Miyato et al., 2018; Wang et al., 2022), attention mechanisms (Henry et al., 2020; Wortsman et al., 2024), model structures (Bachlechner et al., 2021; Qi et al., 2023b), and initialization strategies (Glorot & Bengio, 2010; He et al., 2015; Qi et al.,

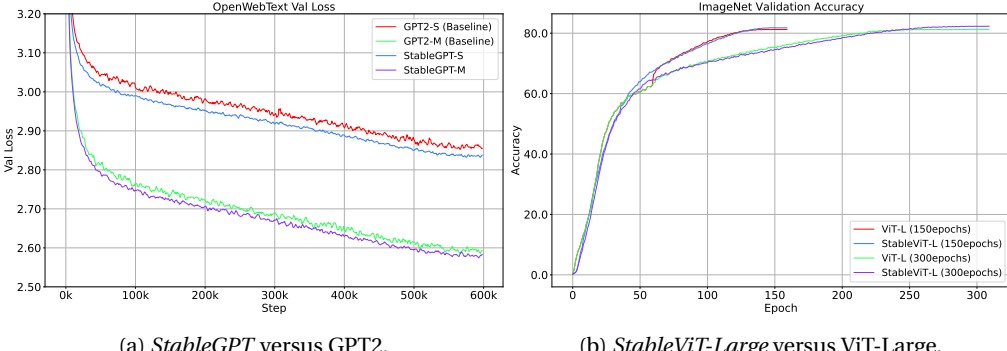

(a) *StableGPT* versus GPT2.

(b) *StableViT-Large* versus ViT-Large.

FIGURE 1: Except for being more stable during the training, *StableGPT* (left) also achieves a better validation loss (**2.827** for *StableGPT-S* (124M) versus **2.848** for GPT2-S (124M), and **2.569** for *StableGPT-M* (350M) versus **2.579** for GPT2-M (350M)), and *StableViT*-L (right) achieves a better recognition accuracy (**82.4% versus 81.3%**) compared to ViT-L under the settings of training 150 or 300 epochs. *StableGPT-S* can get an even better result (validation loss (**2.819**) with a higher learning rate (see Appendix I.), but here for fairness, we keep all the parameters of the optimizer in training are the same as the baseline.

2023b). From the perspective of normalization, it has shown that normalization play a critical role in stabilizing the training of Transformer, *e.g.*, Xiong et al. (2020) demonstrated that Pre-LayerNorm (Pre-LN) offers greater stability compared to Post-LayerNorm (Post-LN), Wang et al. (2022) proposed a DeepNorm and a depth-specific initialization to stabilize Post-LN. From the perspective of attention, Kim et al. (2021) showed that the standard dot-product attention is not Lipschitz continuous and thus introduced an alternative $L_2$ attention to address the continuous issue, QKNorm (Henry et al., 2020; Dehghani et al., 2023) proposed to normalize the query and key matrices in attention mechanisms to improve the stability of the attention module. From the perspective of initialization, Zhang et al. (2019) introduced a fixed-update initialization (Fixup) to prevent gradient exploding or vanishing at the start of training. This method rescales a standard initialization and enables stable training of residual networks without the need for normalization. Each of these approaches contributes to a better understanding and improvement of Transformer training stability, paving the way for more robust and efficient models. Bachlechner et al. (2021) demonstrated that a simple architectural modification, *i.e.*, gating each residual shortcut with a learnable zero-initialized parameter (ReZero), could significantly stabilize the training of Transformer. Using ReZero, they successfully trained Transformers with up to 120 layers. More recently, Qi et al. (2023b) introduced a novel Transformer architecture, called Lips-Former, which is designed to be Lipschitz continuous (*i.e.*, the gradients are bounded) and has been shown more stable during the training. The Lipschitz continuity allows for certain theoretical guarantees about the model's behavior, which is important in reliability or interpretability.

This paper attempts to provide a theoretical understanding of the components that cause training instability of Transformer. To be specific, our main contributions are highlighted as follows.

- We give a theoretical analysis of the Xavier initialization from the perspective of random matrix theory, showing that the Lipschitz constant of the linear projection associated to the Xavier initialization is bounded by 2. Instead, we present a more stable method for initialization, termed *StableInit* (defined in Eq. 1), for which the Lipschitz constant is bounded by 1.

- We dig into the issue in back-propagation of the normalization by analyzing the Jacobian matrix of the normalization layer and find that the factor $\sqrt{d}$ will affect the gradient flow significantly. As a remedy, we derive a more stable design for the normalization, called *StableNorm* (defined in Eq. 2), in which $d^{\alpha}$ with $\alpha \in [0, 0.5]$ is adopted to replace $\sqrt{d}$ in the normalization layer, and verify that using a smaller $\alpha$ (*e.g.*, 0.475, rather than 0.5) will yield smaller gradients and thus lead to more stable training.

- We present a new stable form of attention, named *StableAtten* (defined in Eq. 3), which is built on our *StableNorm* and has the advantage that the logit of the attention is not directly related to the hidden dimension $d$ and thus is robust to the increase of the model scale.

- By putting together the *StableInit*, *StableNorm* and *StableAtten*, we have a stabilized design for Transformer, termed *Stable-Transformer*. In experiments, for the single-direction generative model *i.e.,* GPT (Radford et al., 2018; 2019; Brown et al., 2020), we compile a *StableGPT*; for a bi-direction attention model *i.e.,* ViT (Dosovitskiy et al., 2020), we compile a *StableViT*. We evaluate *StableGPT* and *StableViT* extensively on large model (1B parameters) and deep model (200 layers) that our proposed *Stable-Transformer* achieves a more stable training process.

The paper is organized as follows. We first introduce our experimental setups, and then present our stabilized design of the modules. For each module, we give our mathematical analysis at first and then show empirical evaluation. There is no an independent section for experiments.

## 2 EXPERIMENTAL METHODOLOGY

We evaluate our stabilized components on ViT (Dosovitskiy et al., 2020) and GPT (Radford et al., 2018; 2019; Brown et al., 2020). For general training setting, by default, we use the optimizer Adam Kingma & Ba (2014) with $\beta_1 = 0.9$, $\beta_2 = 0.95$ and $\epsilon = 10^{-8}$, and the gradient clipping is set to 1. When using weight decay, we follow AdamW (Loshchilov & Hutter, 2019), for which only the weight matrix is enforced to the weight matrix but not the 1-d vector (*e.g.,* $\gamma$ and $\beta$ in LayerNorm) and the scalar. We train all models on GPUs A800 in bfloat16 precision using PyTorch (Paszke et al., 2019). We use a cosine-decay (Loshchilov & Hutter, 2016) schedule from a preset maximum learning rate to a preset minimum learning rate.

**Experimental setups for ViT (Dosovitskiy et al., 2020) and our *StableViT*.** We use timm Wightman (2019) [1]. For ViT model, we use two different scales: ViT-Large (ViT-G) and ViT-Huge (ViT-H). The detailed information about these models are summarized in Table 1. For data augmentation, we use the same data augmentation as Adan (Xie et al., 2024). Thus our results are aligned with the results reported in (Xie et al., 2024).

**Experimental setups for GPT2 and our *StableGPT*.** We use nanoGPT[2] (Karpathy, 2022), which is a simple and fast repository for training and fine-tuning the medium-sized GPTs. The GPT2 is implemented in four versions: GPT2-Small (GPT2-S), GPT2-Medium (GPT2-M), GPT2-Large (GPT2-L) and GPT2-XL. Due to time and computational costs, we only use GPT2-Small (GPT2-S), GPT2-Medium (GPT2-M).

We align our experiments with the original repository, and use the exactly same training setting as nanoGPT. Detailed parameters is listed in Table 3. We reproduce the baseline GPT-2 124M model with the same setup as in nanoGPT, for which the training loss is reduced to 2.848. The learning curve of the loss matches to the original nanoGPT.

It is worth to note that when evaluating a module (or method), we keep all the same but the specific module (or method) for fair comparison. To be more specific, when evaluating each of *StableInit*, *StableNorm* and *StableAtten*, we only replace the corresponding module (or method).

## 3 STABLE-TRANSFORMER AND ITS THEORETICAL JUSTIFICATIONS

In this section, we will present our stabilized initialization method *StableInit*, stabilized normalization module *StableNorm*, and stabilized attention mechanism *StableAtten*, individually. Moreover, we will combine them together to build our *Stable-Transformer*. For each method or module, we start with a justification for the instability issue in training and then provide our stabilized designs with both theoretical justification and empirical evaluation.

### 3.1 STABLE INITIALIZATION

The Xavier initialization is a remarkable technique that significantly enhances the training of neural networks by initializing the weights in a way that maintains the variance of activation across layers, to mitigate the vanishing or exploding gradient problem. In Glorot & Bengio (2010), the Xavier initialization is sorted to two types, *i.e.,* Gaussian distribution and uniform

---

[1]https://github.com/huggingface/pytorch-image-models/tree/main
[2]https://github.com/karpathy/nanoGPT

distribution. The Xavier initialization for $W \in \mathbb{R}^{n_{in} \times n_{out}}$ with Gaussian distribution is defined as: $W_{i,j} \overset{\text{i.i.d.}}{\sim} \mathcal{N}\left(0, \frac{2}{n_{in}+n_{out}}\right)$, where $n_{in}$ and $n_{out}$ denote the dimensions of the input and the output. It is widely used in training modern neural networks and is usually as the default initialization method. Therefore, in the following we will only consider the Xavier initialization with Gaussian distribution when mentioning it.

Now we will analyze the property of the Xavier initialization with Gaussian distribution. To begin with, we would like to introduce a theorem from Random Matrix Theory (RMT) (Wigner, 1955; Tao, 2012; Edelman & Rao, 2005). From RMT, we have the theorem about the singular values of a Gaussian random matrix.

---

**Theorem 1 (Singular Value Bounds of a Gaussian Random Matrix)**
*Let $W \in \mathbb{R}^{m \times n}$ have i.i.d. standard Gaussian entries, i.e., $W_{i,j} \overset{\text{iid}}{\sim} \mathcal{N}(0,1)$. For every $m \geq n$, we have the following inequality*

$$\sqrt{m} - \sqrt{n} \leq \mathbb{E}[\sigma_{\min}(W)] \leq \mathbb{E}[\sigma_{\max}(W)] \leq \sqrt{m} + \sqrt{n},$$

*where $\sigma_{\min}(W)$ and $\sigma_{\max}(W)$ denote the minimal and maximal singular values, respectively.*

---

We provide the proof of Theorem 1 via theory from high-dimensional probability (Vershynin, 2010) in the appendix B. Theorem 1 presents that a random matrix initialized by standard Gaussian distribution $\mathcal{N}(0,1)$, the expectations of its largest and smallest singular values are bounded. The expectation of its largest singular value is no more than $\sqrt{m} + \sqrt{n}$, and the expectation of its smallest singular value is no less than $\sqrt{m} - \sqrt{n}$.

According to Theorem 1, and the definition of Xavier initialization, we have the following lemma.

---

**Lemma 1 (Upper Bound of Weight Matrix Norm of Xavier Initialization)**
*Let $W \in \mathbb{R}^{n_{in} \times n_{out}}$ have i.i.d. standard Gaussian entries, i.e., $W_{i,j} \overset{\text{iid}}{\sim} \mathcal{N}(0, \frac{2}{n_{in}+n_{out}})$. we have the following inequality for its maximum singular value, $\mathbb{E}[\sigma_{\max}(W)] \leq 2$.*

---

We provide a proof of Lemma 1 in Appendix E.

**Remark 1.** Back to the year 2010 when we still do not have ResNet (He et al., 2016) and Batch Normalization (Ioffe & Szegedy, 2015), the Xavier initialization is a remarkable technique that enables the researchers to train a network more than 10 layers. Suppose that we have a MLP with 10 linear layers with ReLU (Nair & Hinton, 2010) between two nearby linear layers, with a softmax in the last linear layer, and using a cross-entropy loss, then the expectation of the largest singular value of each layer is up to 2 and it means that the Lipschitz constant (Fazlyab et al., 2019; Kim et al., 2021; Qi et al., 2023a;b) for each linear layer is 2. Therefore, we can compute the Lipschitz constant of the whole network (assuming that the softmax and the cross-entropy has Lipschitz constant 1) as $2^{10} = 1024$, which is under control.

Although Xavier initialization is a popular initialization method, it still has some issues. One main disadvantage in Xavier initialization is that *it is sensitive to the increase of the network depth.* If the number of layers is 50, then the Lipschitz cosntant of the above-mentioned whole network will be extremely huge. To fix this issue, we present a simple and 1-Lipschitz initialization strategy, which is termed **StableInit**. Precisely, we define it as follows:

$$W_{i,j} \overset{\text{iid}}{\sim} \mathcal{N}\left(0, \left(\frac{1}{\sqrt{n_{in}} + \sqrt{n_{out}}}\right)^2\right), \tag{1}$$

where $n_{in}$ and $n_{out}$ are the dimensions of the input and output of the module, respectively. It is easy to see that with our *StableInit* initialization, we have that $\mathbb{E}[\sigma_{\max}(W)] \leq 1$. Similar to the Xavier initialization, we can also consider to multiply a gain on the weight. The gain can be set to be a smaller value when we use deeper networks. By default, we use 1.0 as the gain.

For clarity, we summarize the following two properties of our *StableInit* initialization.

- **1-Lipschitz constant**: with our *StableInit*, a linear projection module has a Lipschitz constant with expectation 1, other than 2 in the original Xavier initialization.

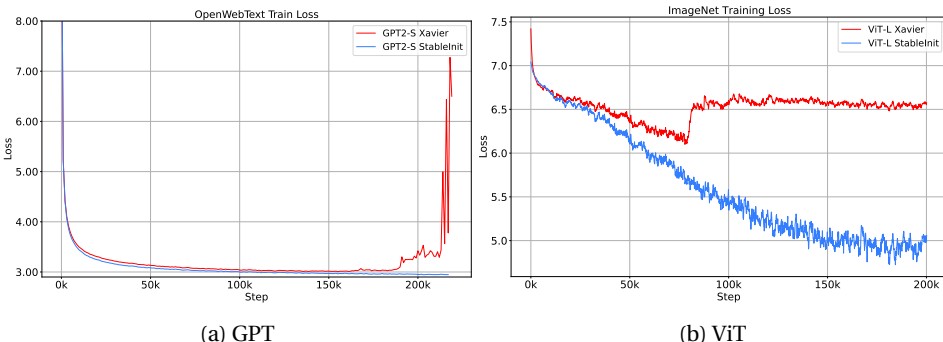

(a) GPT                                          (b) ViT

FIGURE 2: Evaluation of *StableInit* and comparing to the original Xavier initialization. In the legend, "GPT2-S" denotes GPT2-Small. To compare the stability, we do not use learning rate warmup in the evaluation.

- **Less sensitive to depth increase:** our *StableInit* is less sensitive to the depth increase compared to the the original initialization. For a MLP with 10 or 100 linear layers with a ReLU (Nair & Hinton, 2010) between two linear layers, the Lipschitz constant is 1 under our *StableInit*.

### 3.1.1 EVALUATION FOR *StableInit*

These properties of our stabilized Xavier initialization will lead to a more stable training compared to the original Xavier initialization because it has a proper Lipschitz constant.

We can see that from Figure 2 *StableInit* obtains a more stable training property. With *StableInit*, the model can be trained longer until diverge. Theoretically, *StableInit* will be more robust than Xavier initialization for larger model. In experiments, the learning curve of ViT is more jitter because its supervision signal is more sparse, a batch of tokens of GPT is 0.5M tokens, each token will provide a signal, but the batch size 1024 in ViT only provides 1024 supervision signals.

### 3.2 STABLE NORMALIZATION

LayerNorm (Ba et al., 2016) is a technique widely used in deep learning to stabilize and accelerate the training of neural networks. The original definition of LayerNorm is $\text{LN}(\boldsymbol{x}) = \boldsymbol{\gamma} \odot \boldsymbol{z} + \boldsymbol{\beta}$, where $\boldsymbol{z} = \frac{\boldsymbol{y}}{\text{std}(\boldsymbol{y})}$ and $\boldsymbol{y} = \left(\boldsymbol{I} - \frac{1}{d}\mathbf{1}\mathbf{1}^\top\right)\boldsymbol{x}$. After adding a smoothing factor, it can also be written as, $\text{LN}(\boldsymbol{x}) = \boldsymbol{\gamma} \odot \frac{\sqrt{d}\boldsymbol{y}}{\sqrt{\|\boldsymbol{y}\|_2^2 + \epsilon}} + \boldsymbol{\beta}$, and $\boldsymbol{y} = \left(\boldsymbol{I} - \frac{1}{d}\mathbf{1}\mathbf{1}^\top\right)\boldsymbol{x}$, where $\epsilon$ is the smoothing factor, $d$ is the feature dimension of $\boldsymbol{x}$, $\boldsymbol{\gamma}$ and $\boldsymbol{\beta}$ are two learnable $\mathbb{R}^d$ vectors, $\boldsymbol{\gamma}$ and $\boldsymbol{\beta}$ are initialized to 1 and 0. Most recently, some new large language models (Touvron et al., 2023; Chowdhery et al., 2023; Team, 2023) uses RMSNorm (Zhang & Sennrich, 2019) to replace LayerNorm, where RMSNorm is defined as: $\text{RMSN}(\boldsymbol{x}) = \boldsymbol{\gamma} \odot \frac{\sqrt{d}\boldsymbol{x}}{\sqrt{\|\boldsymbol{x}\|_2^2 + \epsilon}}$. Compared to LayerNorm, RMSNorm does not use the bias term and does not conduct the centering.

The Jacobian matrices of LayerNorm and RMSNorm with respect to $\boldsymbol{x}$ are calculated as follows:

$$\frac{\partial \text{LN}(\boldsymbol{x})}{\partial \boldsymbol{x}} = \frac{\partial \boldsymbol{y}}{\partial \boldsymbol{x}} \frac{\partial \text{LN}(\boldsymbol{x})}{\partial \boldsymbol{y}} = \frac{\sqrt{d}}{\sqrt{\|\boldsymbol{y}\|_2^2 + \epsilon}} \left(\boldsymbol{I} - \frac{1}{d}\mathbf{1}\mathbf{1}^\top\right)\left(\boldsymbol{I} - \frac{\boldsymbol{y}\boldsymbol{y}^\top}{\|\boldsymbol{y}\|_2^2 + \epsilon}\right)\text{diag}(\boldsymbol{\gamma}),$$

$$\frac{\partial \text{RMSN}(\boldsymbol{x})}{\partial \boldsymbol{x}} = \frac{\sqrt{d}}{\sqrt{\|\boldsymbol{x}\|_2^2 + \epsilon}} \left(\boldsymbol{I} - \frac{\boldsymbol{x}\boldsymbol{x}^\top}{\|\boldsymbol{x}\|_2^2 + \epsilon}\right)\text{diag}(\boldsymbol{\gamma}).$$

Let us explain each term a little bit individually. It is easy to prove that the maximal singular value of $\left(\boldsymbol{I} - \frac{1}{d}\mathbf{1}\mathbf{1}^\top\right)$ and $\left(\boldsymbol{I} - \frac{\boldsymbol{y}\boldsymbol{y}^\top}{\|\boldsymbol{y}\|_2^2 + \epsilon}\right)$ are both 1. We give a proof of $\sigma_{\max}\left(\boldsymbol{I} - \frac{\boldsymbol{y}\boldsymbol{y}^\top}{\|\boldsymbol{y}\|_2^2 + \epsilon}\right) \leq 1$ in Appendix C. Note that $\sigma_{\max}\left(\boldsymbol{I} - \frac{1}{d}\mathbf{1}\mathbf{1}^\top\right) \leq 1$ is a special case of the former.

To analyze and compare $\frac{\sqrt{d}}{\sqrt{\|\boldsymbol{y}\|_2^2+\epsilon}}$ in $\frac{\partial \mathrm{LN}(\boldsymbol{x})}{\partial \boldsymbol{x}}$ and $\frac{\sqrt{d}}{\sqrt{\|\boldsymbol{x}\|_2^2+\epsilon}}$ in $\frac{\partial \mathrm{RMSN}(\boldsymbol{x})}{\partial \boldsymbol{x}}$, we have the following inequality for the centering transformation.

> **Theorem 2 (Centering Transformation Inequality)**
> *Let $\boldsymbol{y} = \left(\boldsymbol{I} - \frac{1}{d}\mathbf{1}\mathbf{1}^\top\right)\boldsymbol{x}$, we have the following inequality:* $\frac{\sqrt{d}}{\sqrt{\|\boldsymbol{x}\|_2^2+\epsilon}} \le \frac{\sqrt{d}}{\sqrt{\|\boldsymbol{y}\|_2^2+\epsilon}}$.

*Proof.* Centering can be denoted as, $\boldsymbol{y} = \left(I - \frac{1}{d}\mathbf{1}\mathbf{1}^T\right)\boldsymbol{x} = \boldsymbol{x} - \frac{1}{d}\left(\sum_{i=1}^d x_i\right)\mathbf{1}$ Then we have,

$$\|\boldsymbol{y}\|_2^2 = \left(\boldsymbol{x} - \frac{1}{d}\left(\sum_{i=1}^d x_i\right)\mathbf{1}\right)^T \left(\boldsymbol{x} - \frac{1}{d}\left(\sum_{i=1}^d x_i\right)\mathbf{1}\right) = \boldsymbol{x}^\top \boldsymbol{x} - 2\frac{1}{d}\left(\sum_{i=1}^d x_i\right)\mathbf{1}^T\boldsymbol{x} + \frac{1}{d^2}\left(\sum_{i=1}^d x_i\right)^2\mathbf{1}^\top\mathbf{1}$$

Since $\mathbf{1}^\top\boldsymbol{x} = \sum_{i=1}^d x_i$ and $\mathbf{1}^\top\mathbf{1} = d$, we get: $\|\boldsymbol{y}\|_2^2 = \|\boldsymbol{x}\|_2^2 - \frac{1}{d}\left(\sum_{i=1}^d x_i\right)^2$. Since the term $\frac{1}{d}\left(\sum_{i=1}^d x_i\right)^2$ is non-negative, we have $\|\boldsymbol{y}\|_2^2 \le \|\boldsymbol{x}\|_2^2$. Therefore, we have $\frac{\sqrt{d}}{\sqrt{\|\boldsymbol{x}\|_2^2+\epsilon}} \le \frac{\sqrt{d}}{\sqrt{\|\boldsymbol{y}\|_2^2+\epsilon}}$, which proves the inequality. $\qquad\square$

We can see that $\frac{\sqrt{d}}{\sqrt{\|\boldsymbol{y}\|_2^2+\epsilon}}$ in $\frac{\partial \mathrm{LN}(\boldsymbol{x})}{\partial \boldsymbol{x}}$ reaches to the maximum value $\frac{\sqrt{d}}{\sqrt{\epsilon}}$ when std($\boldsymbol{x}$) is 0, but $\frac{\sqrt{d}}{\sqrt{\|\boldsymbol{x}\|_2^2+\epsilon}}$ in $\frac{\partial \mathrm{RMSN}(\boldsymbol{x})}{\partial \boldsymbol{x}}$ reaches to 0 if and only if $\boldsymbol{x}$ is equal to $\mathbf{0}$. Theorem 2 means that RMSNorm is less likely to obtain the maximum value compared to LayerNorm.

We also observe that in the Jacobian matrices of LayerNorm and RMSNorm, both of them have a term $\sqrt{d}$, which is the dimension of $\boldsymbol{x}$ and is also called the hidden dimension of the networks in large language model. With the increase of the hidden dimension $d$ in larger models, there is a square root ratio effect, and thus may lead to larger gradients. Therefore, it will make it harder to train larger models. To alleviate this issue, we present a simple but more stable normalization mechanism as follows:

$$\mathrm{StableNorm}(\boldsymbol{x}) = \boldsymbol{\gamma} \odot \frac{d^\alpha \boldsymbol{x}}{\sqrt{\|\boldsymbol{x}\|_2^2 + \epsilon}}, \tag{2}$$

where $\alpha$ is a hyper-parameter, the range of $\alpha$ is [0, 0.5]. By choosing a reasonable $\alpha$, we can obtain a more stable normalization module. We term our stabilized normalization as **StableNorm**. When $\alpha = 0.5$, *StableNorm* will be equal to RMSNorm (Zhang & Sennrich, 2019).

It is easy to derive that the Jacobian matrix of *StableNorm* is

$$\frac{\partial \mathrm{StableNorm}(\boldsymbol{x})}{\partial \boldsymbol{x}} = \frac{d^\alpha}{\sqrt{\|\boldsymbol{x}\|_2^2 + \epsilon}}\left(\boldsymbol{I} - \frac{\boldsymbol{x}\boldsymbol{x}^\top}{\|\boldsymbol{x}\|_2^2 + \epsilon}\right)\mathrm{diag}(\boldsymbol{\gamma}).$$

We can see that the maximum value is $\frac{d^\alpha}{\sqrt{\epsilon}}$. A reasonable choice for $\epsilon$ is $10^{-5}$. Along with the increase of the hidden dimension in larger model, we can tune the $\alpha$ to make the normalization layer more stable. A good strategy is to choose a smaller $\alpha$ for larger model. To have a more intuitive understanding, let us see an example. Assume $d = 4096$, then $d^\alpha = 64$ when $\alpha = 0.5$ whereas $d^\alpha \approx 42.22$ when $\alpha = 0.45$, it means that the gradient will be scaled by $4096^{0.45} \approx 42.22$ in our *StableNorm* instead of $4096^{0.5} = 64$. Similarly, when $\alpha = 0.4$, the gradient will be scaled by $4096^{0.4} \approx 27.85$ instead of 64 with $\alpha = 0.5$. By choosing 0.45 instead of 0.5, we can actually scale down the gradient by a factor of $\frac{42.22}{64} = 0.66$. Therefore, by choosing a reasonable $\alpha$, our *StableNorm* will help to yield more stable gradients compared to RMSNorm and LayerNorm.

### 3.2.1 EVALUATION FOR *StableNorm*

According to our above derivation, different choices of $\alpha$ in Eq. (2) lead to different Jacobian matrix, and thus lead to different gradient flow. Here, we conduct a set of evaluations with different choices of $\alpha$. We evaluate five different choices of $\alpha$, *i.e.*, $\alpha \in \{0.0, 0.125, 0.25, 0.375, 0.5\}$. We conduct experiments on GPT and ViT, respectively. The empirical results are shown in Figure 3.

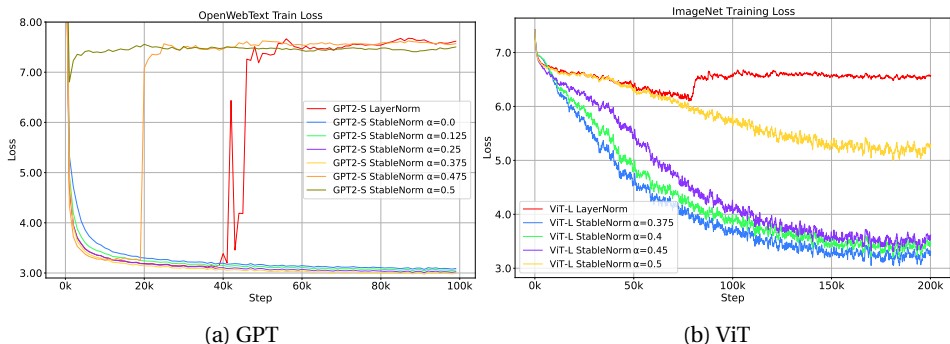

(a) GPT          (b) ViT

FIGURE 3: Evaluation of *StableNorm* and comparison to the original LayerNorm and RM-SNorm on ViT and GPT, respectively. For instance, "StalbeGPT-S $\alpha = 0.5$" in the legend denotes StableGPT-Small model with $\alpha$ set to be 0.5. Where using $\alpha = 0.5$ is reduced to RMSNorm. To compare the stability, we do not use learning rate warmup.

we can see that from Figure 3, choosing an extremely small $\alpha$ may lead to gradient vanishing issue, but choosing a large $\alpha$ may causing training instability. How to choose a good $\alpha$ is an empirical trick. We note here that some relatively large $\alpha$ can be selected for GPT and some relatively small $\alpha$ can be selected for ViT, which may be related to the density and sparsity of the supervised signal of GPT and ViT. When $d$ is large in some large model, a reasonable choice is to choose a smaller $\alpha$ (which takes a lot of resources to verify this. We thus do not do it in this paper). According to our current experiments, we can find that $\alpha$ is a good parameter for balancing gradient vanishing and exploding.

### 3.3 STABLE ATTENTION

Before we present our stabilized attention module, we review the self-attention (Vaswani et al., 2017) and the self-attention with Query-Key normalization (QKNorm) (Henry et al., 2020; Dehghani et al., 2023) at first. Rather than showing the QKNorm works as in Wortsman et al. (2024), we focus on revealing the underlying theoretical reasons. Then we will present our stabilized module, termed *StableAtten*.

#### 3.3.1 WHY SELF-ATTENTION WITH QKNORM WORKS?

For an input sequence $\boldsymbol{X} \in \mathcal{R}^{d \times l}$, $d$ is the dimension of the feature and $l$ is the sequence length, self-attention (Vaswani et al., 2017) is defined as:

$$\boldsymbol{Y} = \boldsymbol{W}_v \boldsymbol{X} \boldsymbol{A}, \quad \boldsymbol{A} = \text{softmax}(\boldsymbol{P}^1), \quad \text{where } \boldsymbol{P}^{(1)} = \frac{\boldsymbol{X}^\top \boldsymbol{W}_q^\top \boldsymbol{W}_k \boldsymbol{X}}{\sqrt{d_1}},$$

in which $d_1 = d/h$ and $h$ is the number of heads, $d_1$ is called as the head dimension, $\boldsymbol{W}_q, \boldsymbol{W}_k \in \mathcal{R}^{d_1 \times d}$, $\boldsymbol{W}_v \in \mathcal{R}^{d_2 \times d}$ (in practice, we usually set $d_1 = d_2$). The size of the output $\boldsymbol{Y} \in \mathcal{R}^{d_2 \times l}$, and the attention matrix $\boldsymbol{A} \in \mathcal{R}^{l \times l}$. Therefore, we have $P_{ij}^{(1)} = \frac{\boldsymbol{x}_i^\top \boldsymbol{W}_q^\top \boldsymbol{W}_k \boldsymbol{x}_j}{\sqrt{d_1}}$ where $\boldsymbol{P}^{(1)}$ is called the logit in (Dehghani et al., 2023; Wortsman et al., 2024).

Self-attention with QKNorm (Dehghani et al., 2023) is defined as:

$$\boldsymbol{A} = \text{softmax}(\boldsymbol{P}^{(2)}), \quad \text{where } P_{ij}^{(2)} = \left( \frac{\text{RMSN}(\boldsymbol{W}_q \boldsymbol{x}_i)^\top \text{RMSN}(\boldsymbol{W}_k \boldsymbol{x}_j)}{\sqrt{d_1}} \right).$$

Note that Dehghani et al. (2023) use a LayerNorm layer without bias and centering, which is equivalent to a RMSNorm Zhang & Sennrich (2019) layer as defined above.

For clarity, we write down the formula for the $i$-th query $\boldsymbol{q}_i$ and the $j$-th key $\boldsymbol{k}_j$ in QKNorm (Dehghani et al., 2023) as follows:

$$\boldsymbol{q}_i = \text{RMSN}(\boldsymbol{W}_q \boldsymbol{x}_i) = \boldsymbol{\gamma}_q \odot \frac{\sqrt{d_1}\boldsymbol{W}_q \boldsymbol{x}_i}{\sqrt{\|\boldsymbol{W}_q \boldsymbol{x}_i\|_2^2 + \epsilon}} = \sqrt{d_1}\,\text{diag}(\boldsymbol{\gamma}_q)\frac{\boldsymbol{W}_q \boldsymbol{x}_i}{\sqrt{\|\boldsymbol{W}_q \boldsymbol{x}_i\|_2^2 + \epsilon}},$$

$$\boldsymbol{k}_j = \text{RMSN}(\boldsymbol{W}_k \boldsymbol{x}_j) = \boldsymbol{\gamma}_k \odot \frac{\sqrt{d_1}\boldsymbol{W}_k \boldsymbol{x}_j}{\sqrt{\|\boldsymbol{W}_k \boldsymbol{x}_j\|_2^2 + \epsilon}} = \sqrt{d_1}\,\text{diag}(\boldsymbol{\gamma}_k)\frac{\boldsymbol{W}_k \boldsymbol{x}_j}{\sqrt{\|\boldsymbol{W}_k \boldsymbol{x}_j\|_2^2 + \epsilon}}.$$

It is easy to see that

$$P_{ij}^{(2)} = \frac{\boldsymbol{q}_i^\top \boldsymbol{k}_j}{\sqrt{d_1}} = \frac{\sqrt{d_1}\sqrt{d_1}\left(\frac{\boldsymbol{W}_q \boldsymbol{x}_i}{\sqrt{\|\boldsymbol{W}_q \boldsymbol{x}_i\|_2^2 + \epsilon}}\right)^\top (\text{diag}(\boldsymbol{\gamma}_q))^\top \text{diag}(\boldsymbol{\gamma}_k)\frac{\boldsymbol{W}_k \boldsymbol{x}_j}{\sqrt{\|\boldsymbol{W}_k \boldsymbol{x}_j\|_2^2 + \epsilon}}}{\sqrt{d_1}}$$

$$= \sqrt{d_1}\left(\frac{\boldsymbol{W}_q \boldsymbol{x}_i}{\sqrt{\|\boldsymbol{W}_q \boldsymbol{x}_i\|_2^2 + \epsilon}}\right)^\top \text{diag}(\boldsymbol{\gamma}_q)\,\text{diag}(\boldsymbol{\gamma}_k)\frac{\boldsymbol{W}_k \boldsymbol{x}_j}{\sqrt{\|\boldsymbol{W}_k \boldsymbol{x}_j\|_2^2 + \epsilon}}.$$

We find that when using QKNorm in self-attention, both the logit and the gradients are *upper bounded*. Precisely, we have the following theorem.

---

**Theorem 3 (Logit Inequality for Self-attention with QKNorm)**
*The logit in self-attention with QKNorm, where $\boldsymbol{\gamma}_q$ and $\boldsymbol{\gamma}_k$ are initialized to 1 and not learned, is upper bounded, i.e., $|P_{ij}^{(2)}| < \sqrt{d_1}$.*

---

*Proof.* We have that when $\boldsymbol{\gamma}_q$ and $\boldsymbol{\gamma}_k$ are initialized to 1, then $P_{ij}^{(2)} = \sqrt{d_1}\left(\frac{\boldsymbol{W}_q \boldsymbol{x}_i}{\sqrt{\|\boldsymbol{W}_q \boldsymbol{x}_i\|_2^2 + \epsilon}}\right)^\top \frac{\boldsymbol{W}_k \boldsymbol{x}_j}{\sqrt{\|\boldsymbol{W}_k \boldsymbol{x}_j\|_2^2 + \epsilon}} = \sqrt{d_1}\,cos(\theta) \le \sqrt{d_1}$, where $\theta$ is the angle between $\frac{\boldsymbol{W}_q \boldsymbol{x}_i}{\sqrt{\|\boldsymbol{W}_q \boldsymbol{x}_i\|_2^2 + \epsilon}}$ and $\frac{\boldsymbol{W}_k \boldsymbol{x}_j}{\sqrt{\|\boldsymbol{W}_k \boldsymbol{x}_j\|_2^2 + \epsilon}}$. $\square$

On contrary, the logit $P_{ij}^{(1)}$ in the original self-attention is not bounded, since that $P_{ij}^{(1)} = \frac{\boldsymbol{x}_i^\top \boldsymbol{W}_q^\top \boldsymbol{W}_k \boldsymbol{x}_j}{d_1}$, where $\boldsymbol{x}_i$, $\boldsymbol{x}_j$ and $\boldsymbol{W}_q$ or $\boldsymbol{W}_k$ might be not bounded. The *upper bounded logit in self-attention with QKNorm* is one of the theoretical reasons that *QKNorm* leads stable training process. Under a fixed hidden dimension $d$, we see that increasing the number of heads $h$ will correspond to decrease the head dimension $d_1$. According to the upper bound, we have that a smaller $d_1$ will stabilize the training process. To verify it, we evaluate the influence of the number of heads in experiment, and show the results in Figure 4. We can observe that increasing the number of heads in the attention does stabilize the training process of the original Transformer.

On the other hand, we find that the gradients in self-attention with QKNorm is also upper bounded. Note that all modules are updated by error back-propagation (Rumelhart et al., 1986; LeCun et al., 2002; 1989; 1998), and computing the gradients in chain is the key. The gradients is self-attention without or with QKNorm can be calculated as follows:

$$\frac{\partial P_{ij}^{(1)}}{\partial \boldsymbol{x}_i} = \boldsymbol{W}_q^\top \boldsymbol{W}_k \boldsymbol{x}_j, \quad \frac{\partial P_{ij}^{(1)}}{\partial \boldsymbol{x}_j} = \boldsymbol{W}_k^\top \boldsymbol{W}_q \boldsymbol{x}_i, \quad \frac{\partial P_{ij}^{(1)}}{\partial \boldsymbol{W}_q} = \boldsymbol{x}_i \boldsymbol{x}_j^\top \boldsymbol{W}_k^\top, \quad \frac{\partial P_{ij}^{(1)}}{\partial \boldsymbol{W}_k} = \boldsymbol{W}_q^\top \boldsymbol{x}_i^\top \boldsymbol{x}_j.$$

and

$$\frac{\partial P_{ij}^{(2)}}{\partial \boldsymbol{x}_i} = \frac{\sqrt{d_1}}{\sqrt{\|\boldsymbol{W}_q \boldsymbol{x}_i\|_2^2 + \epsilon}}\boldsymbol{W}_q^\top \left(\boldsymbol{I} - \frac{\boldsymbol{W}_q \boldsymbol{x}_i \left(\boldsymbol{W}_q \boldsymbol{x}_i\right)^\top}{\|\boldsymbol{W}_q \boldsymbol{x}_i\|_2^2 + \epsilon}\right)\text{diag}(\boldsymbol{\gamma}_q)\,\text{diag}(\boldsymbol{\gamma}_k)\frac{\boldsymbol{W}_k \boldsymbol{x}_j}{\sqrt{\|\boldsymbol{W}_k \boldsymbol{x}_j\|_2^2 + \epsilon}},$$

$$\frac{\partial P_{ij}^{(2)}}{\partial \boldsymbol{W}_q} = \frac{\sqrt{d_1}}{\sqrt{\|\boldsymbol{W}_q \boldsymbol{x}_i\|_2^2 + \epsilon}}\left(\boldsymbol{I} - \frac{\boldsymbol{W}_q \boldsymbol{x}_i \left(\boldsymbol{W}_q \boldsymbol{x}_i\right)^\top}{\|\boldsymbol{W}_q \boldsymbol{x}_i\|_2^2 + \epsilon}\right)\text{diag}(\boldsymbol{\gamma}_q)\,\text{diag}(\boldsymbol{\gamma}_k)\frac{\boldsymbol{W}_k \boldsymbol{x}_j}{\sqrt{\|\boldsymbol{W}_k \boldsymbol{x}_j\|_2^2 + \epsilon}}\boldsymbol{x}_i^\top.$$

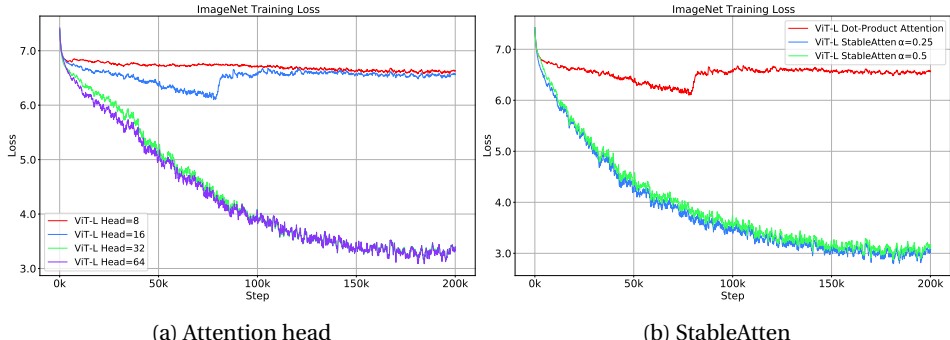

(a) Attention head        (b) StableAtten

FIGURE 4: Evaluation of the number of attention head, and comparison of *StableAtten* with the original dot-product self-attention in Transformer using ViT. For instance, "ViT-L Head=8" in the legend denotes ViT-Large model with 8 attention heads. To compare the training stability, we do not use learning rate warmup in the evaluation.

Let us have a comparison between $\frac{\partial P_{ij}^{(1)}}{\partial \boldsymbol{x}_i}$ and $\frac{\partial P_{ij}^{(2)}}{\partial \boldsymbol{x}_i}$. We have the following observations.

- $\frac{\partial P_{ij}^{(1)}}{\partial \boldsymbol{x}_i}$ is not bounded, but $\frac{\partial P_{ij}^{(2)}}{\partial \boldsymbol{x}_i}$ is bounded by $\frac{\sqrt{d}\,\|\boldsymbol{W}_q\|}{\sqrt{\epsilon}}$. The upper bounded gradients lead to more stable training;

- The value of $\frac{\partial P_{ij}^{(1)}}{\partial \boldsymbol{x}_i}$ is proportion to $\mathcal{O}(\|\boldsymbol{W}_q^\top \boldsymbol{W}_k\|)$, but $\frac{\partial P_{ij}^{(2)}}{\partial \boldsymbol{x}_i}$ is only proportion to $\mathcal{O}(\|\boldsymbol{W}_q\|)$. Generally speaking, the spectral norm of $\mathcal{O}(\|\boldsymbol{W}\|)$ will increase along with the training process and will saturate and oscillate when training comes to convergence.

We also notice of a risk in QKNorm, *i.e.*, there is a factor $\sqrt{d}$ in both $P_{ij}^{(2)}$ and $\frac{\partial P_{ij}^{(2)}}{\partial \boldsymbol{x}_i}$. Therefore, with the increase of model size, there is still some potential risk of instability in training.

### 3.3.2 *StableAtten*

Based on the above analysis, we present a stabilized attention mechanism, called *StableAtten* as,

$$\boldsymbol{A} = \mathrm{softmax}(\tau \boldsymbol{P}^{(3)}), \quad \text{where } P_{ij}^{(3)} = \left( \frac{SN(\boldsymbol{W}_q \boldsymbol{x}_i)^\top SN(\boldsymbol{W}_k \boldsymbol{x}_j)}{d_1^{2\alpha}} \right), \tag{3}$$

where $\tau$ is a temperature coefficient, and $SN(\cdot)$ is a *StableNorm* parameterized by $\alpha$. Since that $\tau$ is likely related to the sequence length $N$, other than the head dimension $d_1$, thus we set $\tau = 1.618 \cdot \log_2(N)$. When the input sequence length is 512, $\tau = 14.562$. We have that

$$P_{ij}^{(3)} = \left( \frac{\boldsymbol{W}_q \boldsymbol{x}_i}{\sqrt{\|\boldsymbol{W}_q \boldsymbol{x}_i\|_2^2 + \epsilon}} \right)^\top \mathrm{diag}(\boldsymbol{\gamma}_q)\, \mathrm{diag}(\boldsymbol{\gamma}_k) \frac{\boldsymbol{W}_k \boldsymbol{x}_j}{\sqrt{\|\boldsymbol{W}_k \boldsymbol{x}_j\|_2^2 + \epsilon}}.$$

The advantage of our stabilized form is that the logit is no longer directly related to the hidden dimension $d$. Although $d$ is large in big model, the range of the logit in our *StableAtten* will not increase with $d$. Each *StableNorm* in query and key normalization will bring in a $d^\alpha$, by dividing $d^{2\alpha}$, we can remove the influence of $d$.

### 3.3.3 EVALUATION FOR *StableAtten*

Here we will evaluate our *StableAtten*, as defined in Eq. 3, and compare it with the original dot-product attention. We evaluate on ViT-Large. The results are shown in Figure 4. We can see that from Figure 4:

- From the left panel of Figure 4, we see that increasing the number of heads in attention will improve the stability of the model training;

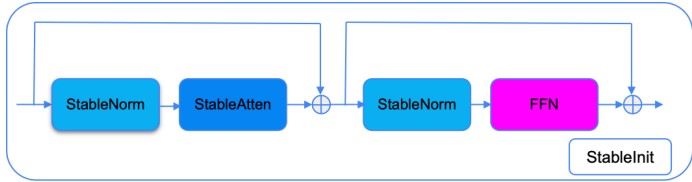

FIGURE 5: One block of our Stable Transformer. Our Stable Transformer uses StableNorm to replace LayerNorm or RMSNorm, use StableAtten to replace the original dot-product attention, and use StableInit to initialize the weights.

- From the right panel of Figure 4, we see that ViT-Large with *StableAtten* are more stable than their corresponding counterparts.

### 3.4 STABLE-TRANSFORMER

Following the architecture of Transformer, we can build a stabilized Transformer. As shown in Figure 5, we use *StableNorm* to replace LayerNorm or RMSNorm, use *StableAtten* to replace the original dot-product attention, and keep the same FFN module. Moreover, we use *StableInit* to initialize the weights.

In practice, our *Stable-Transformer* can lead to two different architectures, *i.e.*, a pure encoder architecture, *i.e.*, Vision Transformer, and a pure decoder architecture, *i.e.*, GPT. Accordingly, we name them as *StableViT* and *StableGPT*, respectively.

To be more specific, for ViT, we build four variants of *StableViT* in correspondence with ViT as detailed in Table 1 in the appendix: StableViT-Large, StableViT-Huge, StableViT-giant, and StableViT-200 (which scales of parameters range from 307M to 1.44B); for GPT, we build three variants of *StableGPT* in correspondence with GPT as detailed in Table 2 in the appendix: StableGPT-Small, StableGPT-Medium and StableGPT-Large (which scales of parameters range from 124M to 774M).

#### 3.4.1 EVALUATION FOR *Stable-Transformer*

To verify the effectiveness of our stabilized architecture, we evaluate different variants of *Stable-ViT* and *Stable-GPT*. The experimental configurations of *StableViT* and *StableGPT* are shown in Table 3 and the experimental results are shown in Figure 1. We find that: our *StableGPT* can be trained smoothly, achieving a better validation loss (*i.e.*, 2.827 versus 2.848) compared to the original GPT2; our *StableViT* yields a better recognition accuracy (*i.e.*, 82.4% versus 81.3%) compared to the original ViT.

Moreover, we note that our *StableViT* and *StableGPT* can also tolerate larger learning rate. More empirical results and details are provided in Appendix I.

## 4 CONCLUSION

We have presented a theoretical analysis for the initialization, normalization and attention module of Transformer from the perspective of training stability. Specifically, we derived an upper bound and a lower bound for the expectations of the maximum and the minimum of the singular values of weight matrix obtained from Xavier initialization, found the reason why increasing hidden dimension can make the normalization layer likely leading to training instability from the Lipschits constant of the Jacobian matrix of the normalization layer, and also pointed out the theoretical mechanism why the hidden dimension can bring instability issue to affect self-attention module. Accordingly, we proposed three stabilized counterpart designs, *i.e.*, StableInit, *StableNorm* and *StableAtten*, and by putting them together, we also proposed a *Stable-Transformer*. We compiled our stabilized components and *Stable-Transformer* with GPT and ViT, and demonstrated that our stabilized methods can improve the training stability, leading improved performance. We hope that our work can benefit the deployment of larger deep models, especially the large language models, in varied application scenarios.

ETHICS STATEMENT

In this paper, we aim to provide a stabilized transformer. Our work does not involve any human subjects, and we have carefully ensured that it poses no potential risks or harms. Additionally, there are no conflicts of interest, sponsorship concerns, or issues related to discrimination, bias, or fairness associated with this study. We have taken steps to address privacy and security concerns, and all data used comply with legal and ethical standards. Our work fully adheres to research integrity principles, and no ethical concerns have arisen during the course of this study.

REPRODUCIBILITY STATEMENT

To ensure the reproducibility of our work, we provide all the details to reproduce the experiments. Theoretical proofs of the claims made in this paper, and detailed experimental settings and configurations are provided in Appendices.

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

## A  NOTATIONS

We primarily follow the notations used in the renowned deep learning book (Goodfellow et al., 2016). We use **bold symbol** to denote a column vector or a matrix, and use non-bold symbol to denote scalar. For instance $y = Wx$, where $x$ and $y$ are two column vectors and $W$ is a projection matrix. We use denominator layout [3], the Jacobian matrix of $y$ with respect to $x$ is $\frac{\partial Wx}{\partial x} = W^\top$, and we have $\frac{\partial x^\top Wx}{\partial x} = (W + W^\top)x$. Using the denominator layout, for a chain function $o = f(g(h(x)))$, where $y = h(x)$, $z = g(y)$, and $o = f(z)$. we have the Jacobian matrix of $o$ with respect to $x$ as $\frac{\partial o}{\partial x} = \frac{\partial y}{\partial x}\frac{\partial z}{\partial y}\frac{\partial o}{\partial z}$.

## B  PROOF OF THEOREM 1

Our proof of Theorem 1 is based on references (Vershynin, 2018; 2010; Mei, Spring 2022). Let us first clarify our problem, we have $W \in \mathcal{R}^{m \times n}$, where $W_{i,j} \overset{\text{iid}}{\sim} \mathcal{N}(0,1)$, we need to prove $\sqrt{m} - \sqrt{n} \leq \mathbb{E}[\sigma_{\min}(W)] \leq \mathbb{E}[\sigma_{\max}(W)] \leq \sqrt{m} + \sqrt{n}$. To prove Theorem 1, we need to prove two parts, *i.e.*, $\mathbb{E}[\sigma_{\max}(W)] \leq \sqrt{m} + \sqrt{n}$ and $\sqrt{m} - \sqrt{n} \leq \mathbb{E}[\sigma_{\min}(W)]$. To prove the first part, we need to first introduce Sudakov-Fernique inequality.

---

**Theorem 4 (Sudakov-Fernique inequality.)**
*Let $(A_{s,t})_{s\in S, t\in T}$ and $(B_{s,t})_{s\in S, t\in T}$ be two zero mean Gaussian processes. Assume that for all $s_1, s_2 \in S$ and $t_1, t_2 \in T$, we have*

$$\mathbb{E}\left[(A_{t_1,s_1} - A_{t_2,s_2})^2\right] \leq \mathbb{E}\left[(B_{t_1,s_1} - B_{t_2,s_2})^2\right].$$

*Then we have*

$$\mathbb{E}\left[\sup_{s\in S, t\in T} A_{s,t}\right] \leq \mathbb{E}\left[\sup_{s\in S, t\in T} B_{s,t}\right].$$

---

Here, we do not provide the proof of Theorem 4. You can find the proof in (Vershynin, 2018). The Sudakov-Fernique inequality will be used in our following proof of $\mathbb{E}[\sigma_{\max}(W)] \leq \sqrt{m} + \sqrt{n}$.

Let us define $A_{u,v} = \langle Wu, v\rangle = v^\top W u$ for $u \in S^{n-1}$ and $v \in S^{m-1}$, where $W_{i,j} \overset{\text{iid}}{\sim} \mathcal{N}(0,1)$. We define

$$B_{u,v} = \langle u, g\rangle + \langle v, h\rangle = \sum_{i=1}^{n} u_i g_i + \sum_{j=1}^{m} v_j g_j, \quad g_i \overset{\text{iid}}{\sim} \mathcal{N}(0,1), h_j \overset{\text{iid}}{\sim} \mathcal{N}(0,1).$$

For any $(u, v), (q, z) \in (S^{n-1} \times S^{m-1})$, let us consider that:

$$\mathbb{E}\left[(A_{u,v} - A_{q,z})^2\right] = \mathbb{E}\left[(\langle Wu, v\rangle - \langle Wq, z\rangle)^2\right]$$

$$= \mathbb{E}\left[\left(\sum_{i,j} W_{ij}(u_j v_i - q_j z_i)\right)^2\right]$$

$$= \sum_{i,j}(u_j v_i - q_j z_i)^2 \quad \text{(by independence, } W_{i,j} \overset{\text{iid}}{\sim} \mathcal{N}(0,1))$$

$$= \|uv^\top - qz^\top\|_F^2$$

$$\leq \|u - q\|_2^2 + \|v - z\|_2^2. \quad \text{(see the following proof.)}$$

We need to prove the following inequality:

$$\|uv^\top - qz^\top\|_F^2 \leq \|u - q\|_2^2 + \|v - z\|_2^2,$$

where $u, q \in S^{n-1}$ and $v, z \in S^{m-1}$, and $S^{n-1}$ denotes the unit sphere in $\mathbb{R}^n$.

---
[3] https://en.wikipedia.org/wiki/Matrix_calculus

*Proof.* First, recall that $\|\boldsymbol{W}\|_F^2 = \operatorname{tr}\left[\boldsymbol{W}^\top \boldsymbol{W}\right]$, we have:

$$\|\boldsymbol{u}\boldsymbol{v}^\top - \boldsymbol{q}\boldsymbol{z}^\top\|_F^2 = \operatorname{tr}\left[(\boldsymbol{u}\boldsymbol{v}^\top - \boldsymbol{q}\boldsymbol{z}^\top)^\top (\boldsymbol{u}\boldsymbol{v}^\top - \boldsymbol{q}\boldsymbol{z}^\top)\right].$$
$$= \operatorname{tr}\left[(\boldsymbol{v}\boldsymbol{u}^\top - \boldsymbol{z}\boldsymbol{q}^\top)(\boldsymbol{u}\boldsymbol{v}^\top - \boldsymbol{q}\boldsymbol{z}^\top)\right].$$
$$= \operatorname{tr}\left[\boldsymbol{v}(\boldsymbol{u}^\top \boldsymbol{u})\boldsymbol{v}^\top - \boldsymbol{v}(\boldsymbol{u}^\top \boldsymbol{q})\boldsymbol{z}^\top - \boldsymbol{z}(\boldsymbol{q}^\top \boldsymbol{u})\boldsymbol{v}^\top + \boldsymbol{z}(\boldsymbol{q}^\top \boldsymbol{q})\boldsymbol{z}^\top\right].$$
$$= \operatorname{tr}\left[\boldsymbol{v}\boldsymbol{v}^\top - \boldsymbol{v}\boldsymbol{z}^\top \boldsymbol{q}^\top \boldsymbol{u} - \boldsymbol{z}\boldsymbol{u}^\top \boldsymbol{q}^\top \boldsymbol{v} + \boldsymbol{z}\boldsymbol{z}^\top\right].$$
$$= \operatorname{tr}(\boldsymbol{v}\boldsymbol{v}^\top) + \operatorname{tr}(\boldsymbol{z}\boldsymbol{z}^\top) - \operatorname{tr}(\boldsymbol{v}\boldsymbol{z}^\top \boldsymbol{q}^\top \boldsymbol{u}) - \operatorname{tr}(\boldsymbol{z}\boldsymbol{u}^\top \boldsymbol{q}^\top \boldsymbol{v}).$$

From the definition of the trace, we have:

$$\operatorname{tr}(\boldsymbol{v}\boldsymbol{v}^\top) = \|\boldsymbol{v}\|_2^2, \quad \operatorname{tr}(\boldsymbol{z}\boldsymbol{z}^\top) = \|\boldsymbol{z}\|_2^2, \quad \operatorname{tr}(\boldsymbol{v}\boldsymbol{z}^\top \boldsymbol{q}^\top \boldsymbol{u}) = (\boldsymbol{v}^\top \boldsymbol{z})(\boldsymbol{q}^\top \boldsymbol{u}), \quad \operatorname{tr}(\boldsymbol{z}\boldsymbol{u}^\top \boldsymbol{q}^\top \boldsymbol{v}) = (\boldsymbol{z}^\top \boldsymbol{v})(\boldsymbol{u}^\top \boldsymbol{q}).$$

Thus, we have:
$$\|\boldsymbol{u}\boldsymbol{v}^\top - \boldsymbol{q}\boldsymbol{z}^\top\|_F^2 = \|\boldsymbol{v}\|_2^2 \|\boldsymbol{u}\|_2^2 + \|\boldsymbol{z}\|_2^2 \|\boldsymbol{q}\|_2^2 - 2(\boldsymbol{v}^\top \boldsymbol{z})(\boldsymbol{u}^\top \boldsymbol{q}).$$

Since $\boldsymbol{u}, \boldsymbol{q} \in S^{n-1}$ and $\boldsymbol{v}, \boldsymbol{z} \in S^{m-1}$, we have $\|\boldsymbol{u}\|_2 = \|\boldsymbol{q}\|_2 = 1$ and $\|\boldsymbol{v}\|_2 = \|\boldsymbol{z}\|_2 = 1$, simplifying to:

$$\|\boldsymbol{v}\|_2^2 \|\boldsymbol{u}\|_2^2 + \|\boldsymbol{z}\|_2^2 \|\boldsymbol{q}\|_2^2 - 2(\boldsymbol{v}^\top \boldsymbol{z})(\boldsymbol{u}^\top \boldsymbol{q}) = 1 + 1 - 2(\boldsymbol{v}^\top \boldsymbol{z})(\boldsymbol{u}^\top \boldsymbol{q}).$$
$$= 2 - 2(\boldsymbol{v}^\top \boldsymbol{z})(\boldsymbol{u}^\top \boldsymbol{q}).$$

Now, consider the right-hand side:

$$\|\boldsymbol{u} - \boldsymbol{q}\|_2^2 + \|\boldsymbol{v} - \boldsymbol{z}\|_2^2. = (\|\boldsymbol{u}\|_2^2 - 2\boldsymbol{u}^\top \boldsymbol{q} + \|\boldsymbol{q}\|_2^2) + (\|\boldsymbol{v}\|_2^2 - 2\boldsymbol{v}^\top \boldsymbol{z} + \|\boldsymbol{z}\|_2^2).$$
$$= 1 - 2\boldsymbol{u}^\top \boldsymbol{q} + 1 + 1 - 2\boldsymbol{v}^\top \boldsymbol{z} + 1.$$
$$= 2 - 2(\boldsymbol{u}^\top \boldsymbol{q}) + 2 - 2(\boldsymbol{v}^\top \boldsymbol{z}).$$
$$= 4 - 2(\boldsymbol{u}^\top \boldsymbol{q}) - 2(\boldsymbol{v}^\top \boldsymbol{z}).$$

let us assume $c = \boldsymbol{u}^\top \boldsymbol{q}$ and $d = \boldsymbol{v}^\top \boldsymbol{z}$. Since $\boldsymbol{u}, \boldsymbol{q} \in S^{n-1}$ and $\boldsymbol{v}, \boldsymbol{z} \in S^{m-1}$, we know $c \le 1$, $d \le 1$, and

$$\left(4 - 2(\boldsymbol{u}^\top \boldsymbol{q}) - 2(\boldsymbol{v}^\top \boldsymbol{z})\right) - \left(2 - 2(\boldsymbol{u}^\top \boldsymbol{q})(\boldsymbol{v}^\top \boldsymbol{z})\right) = 2(1 - c)(1 - d) \ge 0.$$

Similarly, we have,

$$\mathbb{E}\left[\left(B_{\boldsymbol{u},\boldsymbol{v}} - B_{\boldsymbol{q},\boldsymbol{z}}\right)^2\right] = \mathbb{E}\left[\left(\langle \boldsymbol{g}, \boldsymbol{u} - \boldsymbol{q}\rangle + \langle \boldsymbol{h}, \boldsymbol{v} - \boldsymbol{z}\rangle\right)^2\right]$$
$$= \mathbb{E}\left[\langle \boldsymbol{g}, \boldsymbol{u} - \boldsymbol{q}\rangle^2\right] + \mathbb{E}\left[\langle \boldsymbol{h}, \boldsymbol{v} - \boldsymbol{z}\rangle^2\right] \quad \text{(by independence, mean 0)}$$
$$= \|\boldsymbol{u} - \boldsymbol{q}\|_2^2 + \|\boldsymbol{v} - \boldsymbol{z}\|_2^2. \quad \text{(since } \boldsymbol{g}, \boldsymbol{h} \text{ are standard normal)}.$$

Thus, we have

$$\mathbb{E}\left[A_{\boldsymbol{u},\boldsymbol{v}} - A_{\boldsymbol{q},\boldsymbol{z}}\right]^2 \le \mathbb{E}\left[\left(B_{\boldsymbol{u},\boldsymbol{v}} - B_{\boldsymbol{q},\boldsymbol{z}}\right)^2\right].$$

Now, applying the Sudakov-Fernique inequality, we have

$$\mathbb{E}\left[\sup_{(\boldsymbol{u},\boldsymbol{v}) \in S^{n-1} \times S^{m-1}} \langle \boldsymbol{W}\boldsymbol{u}, \boldsymbol{v}\rangle\right] \le \mathbb{E}\left[\sup_{(\boldsymbol{u},\boldsymbol{v}) \in S^{n-1} \times S^{m-1}} \left(\langle \boldsymbol{u}, \boldsymbol{g}\rangle + \langle \boldsymbol{v}, \boldsymbol{h}\rangle\right)\right]$$
$$= \mathbb{E}\left[\sup_{(\boldsymbol{u},\boldsymbol{v}) \in S^{n-1} \times S^{m-1}} \langle \boldsymbol{u}, \boldsymbol{g}\rangle\right] + \mathbb{E}\left[\sup_{(\boldsymbol{u},\boldsymbol{v}) \in S^{n-1} \times S^{m-1}} \langle \boldsymbol{v}, \boldsymbol{h}\rangle\right]$$
$$= \mathbb{E}[\|\boldsymbol{g}\|_2] + \mathbb{E}[\|\boldsymbol{h}\|_2]$$
$$\le \mathbb{E}[\|\boldsymbol{g}\|_2^2]^{1/2} + \mathbb{E}[\|\boldsymbol{h}\|_2^2]^{1/2}$$
$$= \sqrt{n} + \sqrt{m}.$$

This completes the proof of the first part of Theorem 1.

$\square$

To prove the second part, we need to introduce Gordon's Inequality.

---

**Theorem 5 (Gordon's Inequality.)**
*Let $(A_{s,t})_{s \in S, t \in T}$ and $(B_{s,t})_{s \in S, t \in T}$ be two Gaussian processes with $\mathbb{E}[A_{s,t}] = \mathbb{E}[B_{s,t}] = 0$, and suppose that*

$$\begin{cases} \mathbb{E}[(A_{s,t_1} - A_{s,t_2})^2] \geq \mathbb{E}[(B_{s,t_1} - B_{s,t_2})^2] & \forall t_1, t_2 \in T, s \in S, \\ \mathbb{E}[(A_{s_1,t_1} - A_{s_2,t_2})^2] \leq \mathbb{E}[(B_{s_1,t_1} - B_{s_2,t_2})^2] & \forall s_1 \neq s_2 \in S, t_1, t_2 \in T. \end{cases}$$

*Then*

$$\mathbb{E}\left[\sup_{s \in S} \inf_{t \in T} A_{s,t}\right] \leq \mathbb{E}\left[\sup_{s \in S} \inf_{t \in T} B_{s,t}\right].$$

---

Same as above, we do not provide the proof of Gordon's ineqaulity. The proof can be found in (Vershynin, 2018). We will directly use it to help our proof of Theorem 1.

*Proof.* Let $B_{u,v} = \langle g, u \rangle + \langle h, v \rangle$. Check that $A_{u,v}$ and $B_{u,v}$ satisfy the conditions in the theorem. Then we have

$$\begin{aligned}
-\mathbb{E}[\sigma_{\min}(W)] &= \mathbb{E}\left[\sup_{v \in S^{n-1}} -\|Wv\|_2\right] \\
&= \mathbb{E}\left[\sup_{v \in S^{n-1}} \inf_{u \in S^{m-1}} \langle u, -Wv \rangle\right] \\
&\leq \mathbb{E}\left[\sup_{v \in S^{n-1}} \inf_{u \in S^{m-1}} \langle g, u \rangle + \langle h, v \rangle\right] \\
&= \mathbb{E}\left[\sup_{v \in S^{n-1}} \langle h, v \rangle\right] + \mathbb{E}\left[\inf_{u \in S^{m-1}} \langle g, u \rangle\right] \quad \text{(since } g, h \text{ are standard normal.)} \\
&= \mathbb{E}[\|h\|_2] - \mathbb{E}[\|g\|_2] \\
&= \sqrt{n} - \sqrt{m}.
\end{aligned}$$

Therefore, we have

$$\mathbb{E}[\sigma_{\min}(W)] \geq \sqrt{m} - \sqrt{n}.$$

This completes the proof of the second part of Theorem 1. $\square$

## C PROOF OF $\sigma_{max}\left(I - \frac{yy^T}{\|y\|_2^2 + \epsilon}\right) \leq 1$

*Proof.* Let $M = \left(I - \frac{yy^T}{\|y\|_2^2 + \epsilon}\right)$, to prove that the maximum singular value of the matrix $M$ is 1, we need to analyze the properties of this matrix.

Let $A = \frac{yy^T}{\|y\|_2^2 + \epsilon}$, the matrix $A$ is a rank-1 matrix with one non-zero eigenvalue. The non-zero eigenvalue is

$$\lambda_A = \frac{\|y\|_2^2}{\|y\|_2^2 + \epsilon}$$

The eigenvalues of $M$ are $1 - \lambda_A$ and 1 with multiplicity $n - 1$:

$$\lambda_M = 1 - \frac{\|y\|_2^2}{\|y\|_2^2 + \epsilon} = \frac{\epsilon}{\|y\|_2^2 + \epsilon}$$

All other eigenvalues are 1. The singular values of $M$ are the absolute values of its eigenvalues:

$$\sigma_1(M) = 1, \quad \text{(with multiplicity } n-1\text{)}$$

$$\sigma_2(M) = \frac{\epsilon}{\|y\|_2^2 + \epsilon}.$$

The maximum singular value of $M$ is the largest eigenvalue in absolute value, which is 1. Thus, the maximum singular value of the matrix $\left(I - \frac{yy^T}{\|y\|_2^2 + \epsilon}\right)$ is 1. $\qquad\square$

## D   QKNorm Derivations

Here, we list all the partial derivations for the QKNorm.

$$\frac{\partial P_{ij}^{(2)}}{\partial x_i} = \frac{\sqrt{d}}{\sqrt{\|W_q x_i\|_2^2 + \epsilon}} W_q^\top \left(I - \frac{W_q x_i (W_q x_i)^\top}{\|W_q x_i\|_2^2 + \epsilon}\right) \operatorname{diag}(\gamma_q) \operatorname{diag}(\gamma_k) \frac{W_k x_j}{\sqrt{\|W_k x_j\|_2^2 + \epsilon}},$$

$$\frac{\partial P_{ij}^{(2)}}{\partial x_j} = \frac{\sqrt{d}}{\sqrt{\|W_k x_j\|_2^2 + \epsilon}} W_k^\top \left(I - \frac{W_k x_j (W_k x_j)^\top}{\|W_k x_j\|_2^2 + \epsilon}\right) \operatorname{diag}(\gamma_k) \operatorname{diag}(\gamma_q) \frac{W_q x_i}{\sqrt{\|W_q x_i\|_2^2 + \epsilon}},$$

$$\frac{\partial P_{ij}^{(2)}}{\partial W_q} = \frac{\sqrt{d}}{\sqrt{\|W_q x_i\|_2^2 + \epsilon}} \left(I - \frac{W_q x_i (W_q x_i)^\top}{\|W_q x_i\|_2^2 + \epsilon}\right) \operatorname{diag}(\gamma_q) \operatorname{diag}(\gamma_k) \frac{W_k x_j}{\sqrt{\|W_k x_j\|_2^2 + \epsilon}} x_i^\top,$$

$$\frac{\partial P_{ij}^{(2)}}{\partial W_k} = \frac{\sqrt{d}}{\sqrt{\|W_k x_j\|_2^2 + \epsilon}} \left(I - \frac{W_k x_j (W_k x_j)^\top}{\|W_k x_j\|_2^2 + \epsilon}\right) \operatorname{diag}(\gamma_k) \operatorname{diag}(\gamma_q) \frac{W_q x_i}{\sqrt{\|W_q x_i\|_2^2 + \epsilon}} x_j^\top.$$

When considering $\gamma_q$ and $\gamma_k$ are set to be 1, $\frac{\partial P_{ij}^{(2)}}{\partial x_i}$ is only proportion to $\mathcal{O}(\|W_q\|)$ and $\frac{\partial P_{ij}^{(2)}}{\partial W_q} \le \frac{\sqrt{d}}{\sqrt{\epsilon}}$.

## E   Proof of Lemma 1

*Proof.* To prove $\mathbb{E}[\sigma_{\max}(W)] \le 2$, it is equivalent to prove $\sqrt{\frac{2}{n_{in} + n_{out}}}(\sqrt{n_{in}} + \sqrt{n_{out}}) \le 2$ for any $n_{in}$ and $n_{out}$. Note that:

$$\left(\sqrt{\frac{2}{n_{in} + n_{out}}}(\sqrt{n_{in}} + \sqrt{n_{out}})\right)^2 = \frac{2(\sqrt{n_{in}} + \sqrt{n_{out}})^2}{n_{in} + n_{out}} = \frac{2(n_{in} + n_{out}) + 4\sqrt{n_{in} n_{out}}}{n_{in} + n_{out}} = 2 + \frac{4\sqrt{n_{in} n_{out}}}{n_{in} + n_{out}} \le 4.$$

Thus, we have $\left(\sqrt{\frac{2}{n_{in} + n_{out}}}(\sqrt{n_{in}} + \sqrt{n_{out}})\right) \le 2$. $\qquad\square$

## F   Model and Training Configuration

**Model Configurations.**  We list some basic configurations of our *StableViT* and *StableGPT* in Table 1 and Table 2.

**Training Configurations.**  We list the training configurations of our *StableGPT* and *StableViT* in Table 3. For *StableGPT*, we fully follow the experimental configurations of nanoGPT (Karpathy, 2022), all parameters are same as GPT2 (Radford et al., 2019). All experiments are conducted on

TABLE 1: Model configuration for *StableViT*. The *StableViT* is similar with the original ViT (Dosovitskiy et al., 2020).

| Model Card | Params. | Blocks | Embed. dim. | MLP. dim. | Heads | Epochs | Peak LR |
|---|---|---|---|---|---|---|---|
| *StableViT*-L-16 | 307M | 24 | 1024 | 4096 | 16 | 150 or 300 | 1e-3 |
| *StableViT*-H-14 | 632M | 32 | 1280 | 5120 | 16 | 150 or 300 | 1e-3 |
| *StableViT*-g-14 | 1011M | 40 | 1408 | 6144 | 16 | 150 or 300 | 1e-3 |
| *StableViT*-200 | 1439M | 200 | 768 | 3072 | 12 | 150 or 300 | 1e-3 |

TABLE 2: Model configuration for *StableGPT*. The *StableGPT* is similar with the original GPT2 (Radford et al., 2019). We do not include larger models as in nanoGPT (Karpathy, 2022) because training larger models will cost much more computational resource.

| Model Card | Params. | Blocks | Embed. dim. | Heads | Train steps | Peak LR | Minimum LR |
|---|---|---|---|---|---|---|---|
| *StableGPT*-S | 124M | 12 | 768 | 12 | 600K | 6e-4 | 6e-5 |
| *StableGPT*-M | 350M | 24 | 1024 | 16 | 600K | 3e-4 | 3e-5 |
| *StableGPT*-L | 774M | 36 | 1280 | 20 | 600K | 2.5e-4 | 2.5e-5 |

A800 GPU cluster. For instance, it takes around 3 days to train *StableGPT*-Small on a GPU server with 8 A800 GPUs. *StableGPT*-Medium will take around 7.5 days. Note that in the original ViT, we use 60 epochs' learning rate warmup, but in our *StableViT*, we do not use warmup. We do not include some new optimizer (Liu et al., 2023) or learning schedule (Defazio et al., 2024) to further improve the performance of the models.

TABLE 3: Training configurations for *StableGPT* and *StableViT*.

(a) Training configurations for *StableGPT*.

| training config | StableGPT-S/M/L |
|---|---|
| weight init | *StableInit* |
| optimizer | AdamW |
| baseline learning rate | 0.0006 |
| weight decay | 0.1 |
| optimizer momentum | $\beta_1, \beta_2 = 0.9, 0.95$ |
| warmup | 2,000 |
| tokens seen each update | 500,000 |
| max iters | 600,000 |
| batch size | 480 |
| sequence length | 1024 |
| dropout | 0.0 |
| bfloat16 | True |
| gradient clipping | 1.0 |

(b) Training configurations for *Stable-ViT*.

| training config | StableViT-L/H/g/200 $(224^2)$ |
|---|---|
| weight init | *StableInit* |
| optimizer | AdamW |
| base learning rate | 1e-3 |
| weight decay | 0.1 |
| optimizer momentum | $\beta_1, \beta_2 = 0.9, 0.99$ |
| batch size | 1024 |
| training epochs | 300 or 150 or 60 |
| learning rate schedule | cosine decay |
| warmup epochs | 0 |
| randaugment | $(9, 0.5)$ |
| mixup | 0.8 |
| cutmix | 1.0 |
| random erasing | 0 |
| label smoothing | 0.1 |
| stochastic depth | 0.5/0.5 |
| gradient clip | None |
| exp. mov. avg. (EMA) | no |

## G  DEMONSTRATION CODE

To help the audience understand the details of the introduced modules, we list our demonstration codes.

CODE 1: Stable-Transformer Implementation Demonstration.

```python
import torch
import torch.nn as nn
import math

def StableInit(module: nn.Module, name: str = '') -> None:
    if isinstance(module, nn.Linear):
        n_in, n_out = module.weight.shape[0], module.weight.shape[1]
        init_std = 1.0/(math.sqrt(n_in)+math.sqrt(n_out))
        torch.nn.init.normal_(module.weight, mean=0.0, std=init_std)
        if module.bias is not None:
            nn.init.zeros_(module.bias)

class StableNorm(nn.Module):
    def __init__(self, ndim: int, alpha: float = 0.0, eps: float = 1e-8):
        super().__init__()
        self.alpha = alpha
        self.ndim = ndim
        self.eps = eps
        self.weight = nn.Parameter(torch.ones(ndim))

    def forward(self, input):
        x_norm = torch.norm(input, dim=2, keepdim=True) + self.eps
        x = math.pow(self.ndim, self.alpha)*input/x_norm
        y = self.weight.unsqueeze(0).unsqueeze(0)*x
        return y

class StableAtten(nn.Module):
    def __init__(self, dim: int, num_heads: int = 8, qkv_bias: bool = False,
            attn_drop: float = 0., proj_drop: float = 0.,
            norm_layer: nn.Module = StableNorm,
            temperature: float = 1.0, sequence_length: int=0) -> None:
        super().__init__()
        assert dim % num_heads == 0,
        self.num_heads = num_heads
        self.head_dim = dim // num_heads
        self.scale = self.head_dim ** -0.5
        self.qkv = nn.Linear(dim, dim * 3, bias=qkv_bias)
        self.q_norm = norm_layer(self.head_dim)
        self.k_norm = norm_layer(self.head_dim)
        norm_alpha = 2 * self.q_norm.alpha
        self.tau = 1.618*math.log(sequence_length,2)*temperature
        self.scale = self.head_dim**(-norm_alpha)*self.tau
        self.attn_drop = nn.Dropout(attn_drop)
        self.proj = nn.Linear(dim, dim)
        self.proj_drop = nn.Dropout(proj_drop)

    def forward(self, x: torch.Tensor) -> torch.Tensor:
        B, N, C = x.shape
        qkv = self.qkv(x).reshape(B,N,3,self.num_heads,self.head_dim)
        qkv = qkv.permute(2,0,3,1,4)
        q, k, v = qkv.unbind(0)
        q, k = self.q_norm(q), self.k_norm(k)
        q = q * self.scale
        attn = q @ k.transpose(-2, -1)
        attn = attn.softmax(dim=-1)
        attn = self.attn_drop(attn)
        x = attn @ v

        x = x.transpose(1, 2).reshape(B, N, C)
        x = self.proj(x)
        x = self.proj_drop(x)
        return x
```

## H  DISCUSSION ABOUT INITIALIZATION IMPLEMENTATION IN NANOGPT

We observe that, in some popular open-sourced project, *e.g.*, nanoGPT, they use an initialization implementation as code below. Let us consider a model with hidden dimension 768. Suppose

we have a linear layer projecting a 768-d feature into a new 768-d feature. For such a linear layer, the used standard variance is math.sqrt($\frac{2}{768+768}$) $\approx$ 0.036. For *StableNorm*, the used standard variance is $\frac{1}{2*\text{math.sqrt}(768)}$ $\approx$ 0.018. In the following code, the used standard variance is 0.02. It works. However, when we train a GPT-3 175B model with hidden dimension 12288, the standard variance 0.02 is too large. For a GPT-3 175B model with hidden dimension 12288, For *StableNorm*, the used standard variance is $\frac{1}{2*\text{math.sqrt}(12768)}$ $\approx$ 0.0045.

CODE 2: Initilization Implementation in nanoGPT.

```
1  def _init_weights(self, module):
2      if isinstance(module, nn.Linear):
3          torch.nn.init.normal_(module.weight, mean=0.0, std=0.02)
4          if module.bias is not None:
5              torch.nn.init.zeros_(module.bias)
6      elif isinstance(module, nn.Embedding):
7          torch.nn.init.normal_(module.weight, mean=0.0, std=0.02)
```

In conclusion, this implementation works for small model, but it will make training unstable or harder to train when the model is large, *e.g.*, GPT-3 13B or GPT-3 175B.

## I ABLATION STUDY

**StableGPT can tolerate larger learning rate.** To further validate the stability of our algorithm, we used larger learning rates (1.2e-3, 1.8e-3, 2.4e-3) to test our model. As shown in Figure 6, we found that our model can tolerate higher learning rates while maintaining good stability. Meanwhile, we can see that StableGPT-S using 1.2e-3 learning rate achieves a better performance than 6e-4 (**2.819 verse 2.827**).

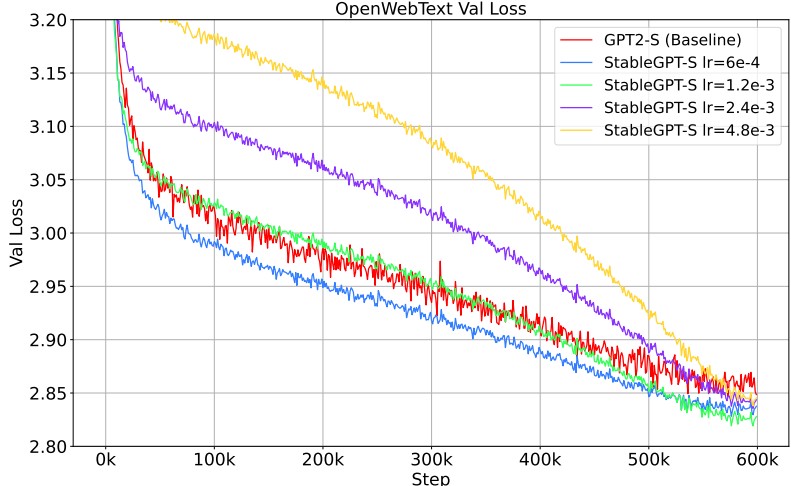

FIGURE 6: StableGPT can tolerate larger learning rate.

**StableGPT is robust to the temperature coefficient in StableAtten.** We conducted an evaluation of the parameter $\tau$ in the *StableAtten*, using values of $\tau = 0.809 \log_2 N$, $\tau = 1.618 \log_2 N$, and $\tau = 3.236 \log_2 N$, the used learning rate here is 6e-4 for all comparisons. We found that our algorithm is relatively robust to this parameter, with performance remaining stable across these values.

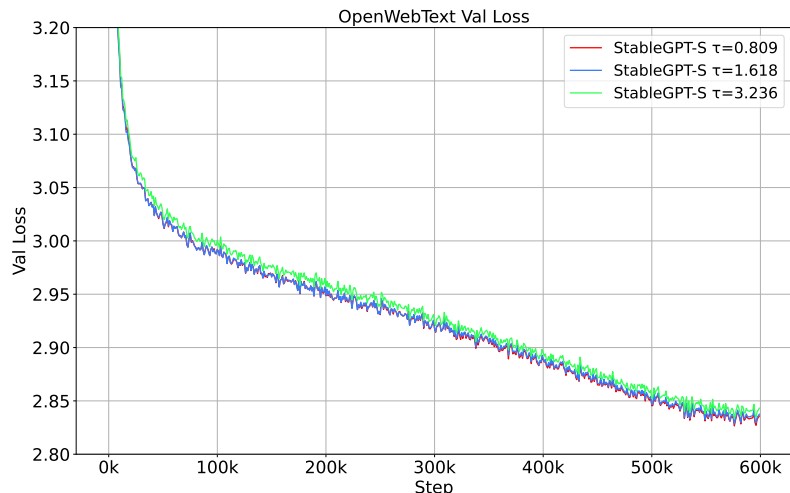

FIGURE 7: Evaluation of temperature coefficient in *StableAtten*.

**About *StableViT*-Huge.** We also conducted evaluations and comparisons on larger *StableViT* models, as shown in Figure 8. Compared to ViT-Huge, our algorithm demonstrates better performance, 81.8 (*StableViT*-Huge) versus 80.5 (ViT-Huge). We also noticed that Model *StableViT*-Huge is not as good as Model *StableViT*-Large, which may be mainly due to two aspects: 1). Insufficient data leading to a certain degree of overfitting, 2). Inadequate data augmentation, although we have adopted data augmentation methods similar to those in previous papers (Xie et al., 2024).

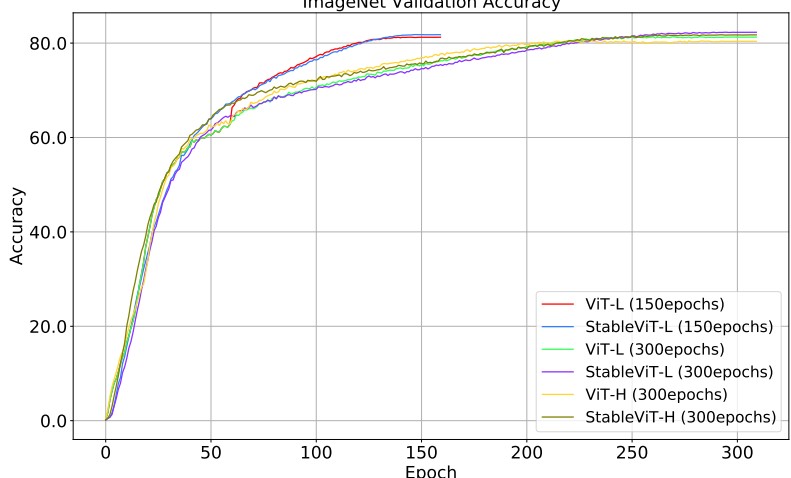

FIGURE 8: Evaluation of *StableViT*-Huge.

## J  EXPERIMENT OF 1B STABLEVIT

To further evaluate the effectiveness of our method at a larger scale, we assessed StableViT with 1B parameters, we term it as StableViT-g where "g" means giant. The model architecture consists of 40 layers with a hidden dimension of 1408, 16 attention heads, and an MLP dimension of 6144. The total parameter count is 1011M, around one billion parameters. We conducted a comparative study between StableViT-g and ViT-g, where ViT-g was evaluated under two settings: with

and without learning rate warmup. Our StableViT-g does not use warmup. In StableViT, we use an $\alpha$ value of 0.25. The comparison results are presented in Figure 9 and Figure 10.

Figure 9 shows that ViT-g crashes after only a few training steps when running without warmup. While the use of warmup enables ViT-g to complete training, our StableViT-g not only achieves stable training without warmup but also demonstrates superior performance. Meanwhile, from Figure 10, we can also observe that the loss of StableViT-g has no spike, but ViT-g even with learning rate warmup has a spike.

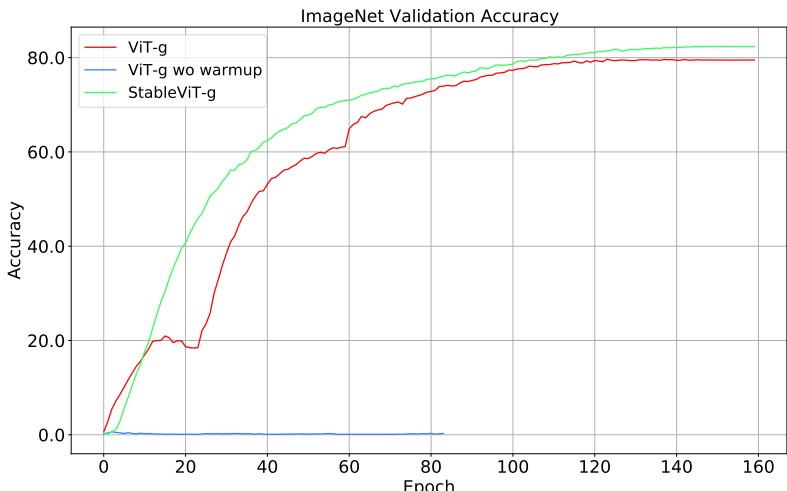

FIGURE 9: Accuracy of *StableViT*-g compared with ViT-g.

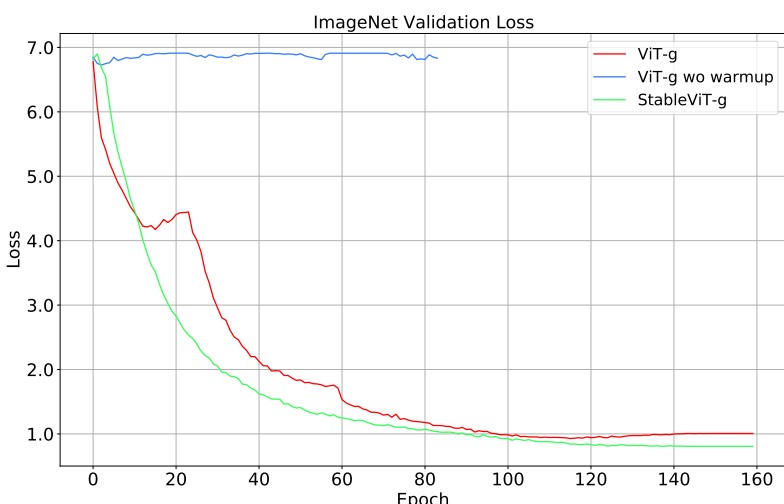

FIGURE 10: Loss curve of *StableViT*-g compared with ViT-g.

## K    EXPERIMENT OF 0.77B STABLEGPT

We also evaluated the effectiveness of StableGPT at a larger scale, termed as StableGPT-large. The model architecture consists of 36 layers with a hidden dimension of 1280 and 20 attention heads. The total parameter count is 774M. Our experimental setup strictly follows the nanoGPT configuration, including all learning rate settings. It is important to note that training StableGPT-large is computationally intensive, requiring two weeks to train 600K steps on 16 A800 GPUs. To reduce the training time, we limited our training to 100K steps instead of the full 600K steps.

The comparison results are presented in Figure 11. We can see from Figure 11, StableGPT-large obtains a better validation loss, 2.523 versus 2.536, than its counterpart, GPT2-large.

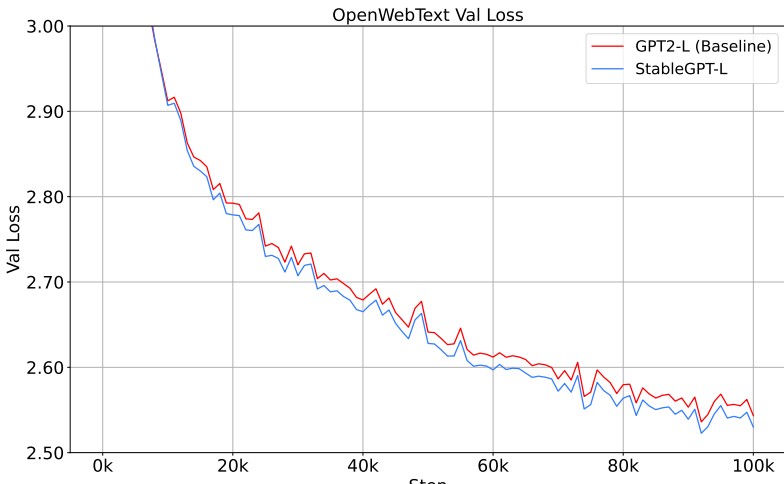

FIGURE 11: Evaluation of *StableGPT*-Large compared with CPT2-Large.

## L  EXPERIMENT OF 200 LAYERS' STABLEVIT WITH 1.44B PARAMETERS

To further verify the stability of our Stable-Transformer, we conduct an experiment of super deep StableViT that has 200 layers. The model architecture consists of 200 layers with a hidden dimension of 768, 12 attention heads, and an MLP dimension of 3072. The total parameter count is 1439M, around 1.4B. We term our model as StableViT-200. Finally, StableViT-200 has 1.44B parameters. The $\alpha$ in StableNorm is set to be 0.25. We compare StableViT-200 with ViT-200.

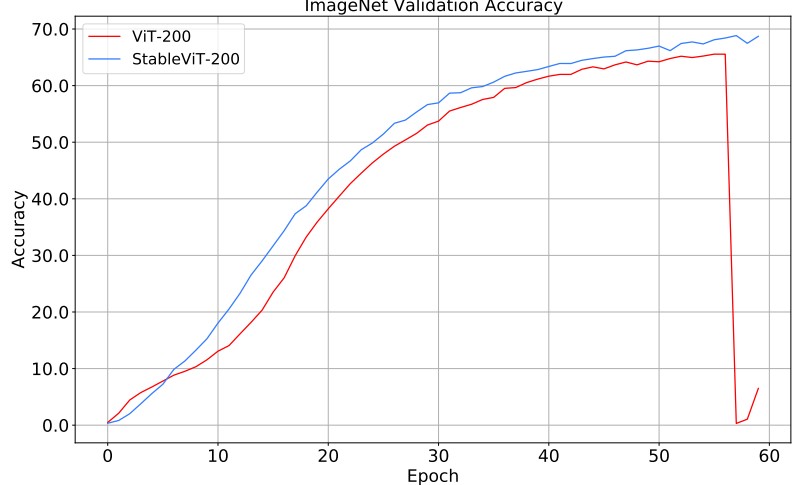

FIGURE 12: Accuracy of *StableGPT*-200 compared with ViT-200.

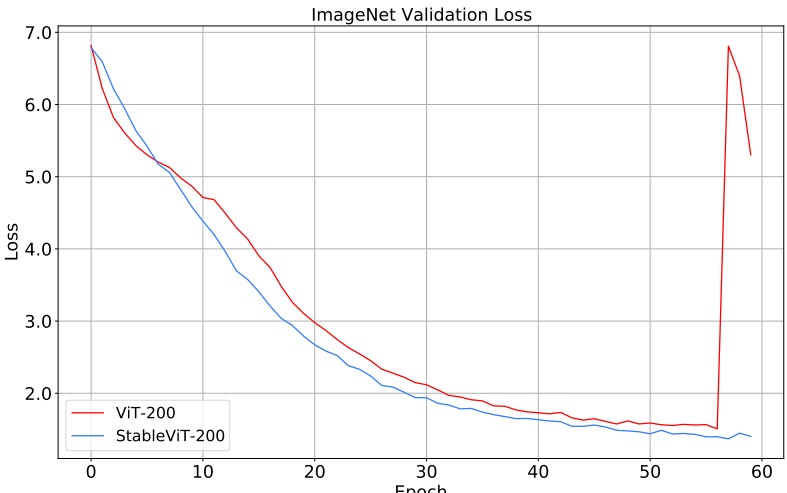

FIGURE 13: Loss curve of *StableGPT*-200 compared with ViT-200.

From Figure 13, with learning rate warmup, ViT-200 has a smoothing loss curve in the early stage, but the loss spikes around at 56th epochs. But StableViT-200, without using learning rate warmup, can converge stably. It fully verify the stability of StableViT in very deep Transformer.

## M  DISCUSSION ABOUT LIPSCHITZ CONSTANT OF STABLENORM

Lipschitz continuity of the network is a very important condition for a stable training. Actually the principle behind StableNorm can also be explained by its Lipschitz constant. Note that the Jacobian matrix of StableNorm is defined as

$$\frac{\partial \text{StableNorm}(\boldsymbol{x})}{\partial \boldsymbol{x}} = \frac{d^{\alpha}}{\sqrt{\|\boldsymbol{x}\|_2^2 + \epsilon}}\left(\boldsymbol{I} - \frac{\boldsymbol{x}\boldsymbol{x}^{\top}}{\|\boldsymbol{x}\|_2^2 + \epsilon}\right)\text{diag}(\boldsymbol{\gamma})$$

and the Jacobian matrix of RMSNorm is defined as

$$\frac{\partial \text{RMSNorm}(\boldsymbol{x})}{\partial \boldsymbol{x}} = \frac{d^{0.5}}{\sqrt{\|\boldsymbol{x}\|_2^2 + \epsilon}}\left(\boldsymbol{I} - \frac{\boldsymbol{x}\boldsymbol{x}^{\top}}{\|\boldsymbol{x}\|_2^2 + \epsilon}\right)\text{diag}(\boldsymbol{\gamma}).$$

By choosing a smaller $\alpha$, *e.g.*, $\alpha < 0.5$, the Lipschitz constant of StableNorm will less than that of RMSNorm. For example, if $d = 1024$, when we choose $\alpha = 0.475$, the Lipschitz constant of StableNorm is only around 84% of that of RMSNorm. This explains why StableNorm has a better stability than RMSNorm.

## N  STABLEATTEN COMPARED WITH $L_2$ SELF-ATTENTION

We further compared our StableAtten with $L_2$ self-attention (Kim et al., 2021). As shown in  (Kim et al., 2021), a necessary condition to guarantee its Lipschitz continuity is $\boldsymbol{W}_q = \boldsymbol{W}_k$, thus we evaluate two versions of $L_2$ self-attentions: a) using tied $\boldsymbol{W}_q$ and $\boldsymbol{W}_k$, *i.e.*, $\boldsymbol{W}_q = \boldsymbol{W}_k$ and b) using two separate $\boldsymbol{W}_q$ and $\boldsymbol{W}_k$, *i.e.*, $\boldsymbol{W}_q \neq \boldsymbol{W}_k$. We conduct experiments to compare the two versions of $L_2$ self-attention methods with StableGPT-large, where the same training settings as the experiments in Appendix K are used, and show in Figure 14 the validation losses of our StableGPT-g and ViT-g with $L_2$ self-attention.

We can see from Figure 14 that, StableGPT-L with StableAtten achieves better validation loss than that of using the $L_2$ self-attention methods. Note that the performance degenerates notably when using the $L_2$ self-attention with tied $\boldsymbol{W}_q$ and $\boldsymbol{W}_k$.

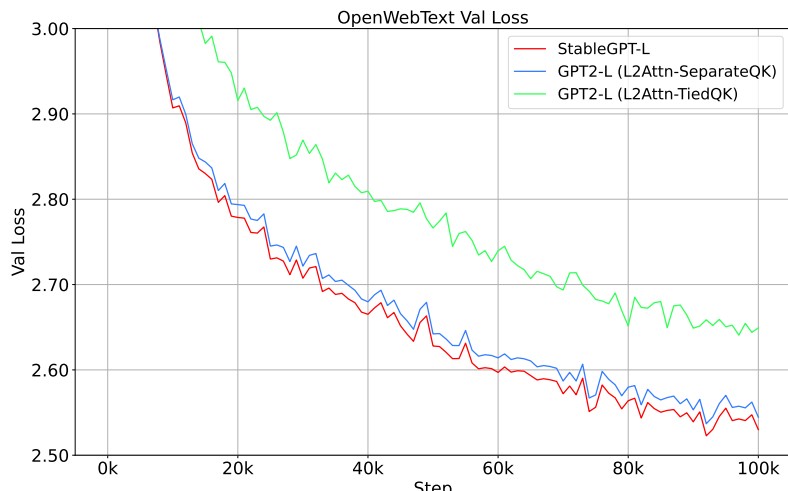

FIGURE 14: The curve of validation loss of *StableGPT*-g compared to ViT-g with $L_2$ self-attention (Kim et al., 2021) under two settings.

## O ROBUSTNESS TO DISTRIBUTION SHIFT

We further conduct a set of experiments to compare the robustness between StableViT-small and ViT-small against to the distribution shift on CIFAR-100. The protocol in experiments is to train both models on the original dataset CIFAR-100 for 200 epochs, with a batch size 512, a learning rate 1e-3, and a weight decay of 1e-4, and then to evaluate the trained models on the original test images of CIFAR-100 and the corrupted test images of CIFAR-100, respectively. Experimental results are reported in Table 4.

TABLE 4: Evaluation (Accuracy) of robustness of StableViT against to distribution shift.

| Models | CIFAR-100 | CIFAR-100-C |
|---|---|---|
| ViT-small | 67.3 | 51.5 |
| StableViT-small | 69.9 | 53.4 |

We can see that from Table 4, StableViT-small obtains a better accuracy than ViT-small, and improves the accuracy from 67.3% to 69.9%. On the corrupted CIFAR-100-C dataset, StableViT-small also shows a better robustness to corruption from 51.5% to 53.4%.

## P RELATED WORK

**Initialization.** Xavier Initialization does the most groundbreaking work in model Initialization. it sets the weights to ensure the variance of activations remains constant across layers, relieving the vanishing and exploding gradient problems. Sutskever et al. (2013) investigates the importance of initialization and momentum (Nesterov, 1983; 1998) in deep learning. Kaiming Initialization (He et al., 2015), builds on Xavier Initialization by scaling the weights for ReLU activations (Nair & Hinton, 2010). Admin (Liu et al., 2020) introduces an adaptive initialization method that dynamically adjusts the initialization parameters based on the network's depth and width. Saxe et al. (2013) introduce an orthogonal initialization, which further optimizes the initial parameter distribution to boost training outcomes. Arpit et al. (2019) also investigates the orthogonal initialization. Huang et al. (2020) propose to scale decoder by $(9L)^{-\frac{1}{4}}$ and scale encoder by $0.67L^{-\frac{1}{4}}$,

this initialization method can be seen as a depth-aware initialization. *Different from the above-mentioned methods, our StableInit is built on Random Matrix Theory, can promise the weight initialized by StableInit has Lipschitz constant approximately 1.*

**Normalization.** LayerNorm (Ba et al., 2016), different from BatchNorm (Ioffe & Szegedy, 2015), normalizes across the features for each data point, making it effective for recurrent and transformer-based architectures. Wang et al. (2019) discuss the influence of Pre-Norm and Post-Norm on the training deep transformer. Xiong et al. (2020) further discuss the influence of pre-norm and post-norm on the training stability. RMSNorm (Zhang & Sennrich, 2019) is a variant of LayerNorm that uses root mean square statistics, offering computational efficiency. DeepNorm (Wang et al., 2022) extends normalization strategies to deep transformer networks. WeightNorm (Salimans & Kingma, 2016) reparameterizes weight vectors to decouple the magnitude from the direction, facilitating smoother optimization. CenterNorm (Qi et al., 2023b) only conducts the centering but does not scaling the feature. ScaleNorm (Nguyen & Salazar, 2019) normalizes only by the scale of the feature vectors, simplifying the normalization process. *RMSNorm and ScaleNorm can be seen as a special case of our StableNorm where $\alpha = 0.5$ and $\alpha = 0$. By choosing a better $\alpha$, our StableNorm can obtain a better training stability.*

**Attention.** Attention mechanism (Bahdanau et al., 2014) is firstly introduced to neural machine translation. Scaled dot-product attention, used in the Transformer architecture, calculates the attention weights using the scaled dot-product of query and key vectors, providing an efficient way to capture dependencies. L2 distance attention employs the Euclidean distance between queries and keys to compute attention scores. Attention with QK-Norm (Henry et al., 2020) normalizes the query and key vectors before computing attention, improving stability and performance. Dehghani et al. (2023) scale the model to 22B via bringing QKNorm into attention. Wortsman et al. (2024) further experimentally evaluate the value of QKNorm on small-scale models. However, these three papers do not mathematically explain why QKNorm works. Liu et al. (2022) introduce to use a Scaled Cosine Attention (SCA) for Transformer. Meanwhile, Qi et al. (2023a) also propose to use scaled cosine similarity attention (SCSA) to compute attention weights. Different from Liu et al. (2022), SCSA (Qi et al., 2023a) multiply a temperature coefficient instead of dividing a temperature coefficient. Cosine similarity attention and attention with QK-Norm share the similar idea, except that the former uses a scalar as a scale, but the latter uses a vector $\gamma$, SCSA also normalizes the values but the latter does not. *StableAtten, the logit of the attention will not be directly related to the hidden dimension $d$, and thus it is robust to the increase of the model scale.*

**Neural Network Stability.** To obtain a better training stability, ReZero (Bachlechner et al., 2021) introduces a simple yet effective mechanism where residual connections start as zero, allowing networks to learn identity mappings more easily and stabilize training. Admin (Liu et al., 2020) not only offers an initialization scheme but also contributes to network stability by dynamically adjusting learning rates and weight decay. DeepNorm (Wang et al., 2022) extends its benefits to network stability by adjusting normalization parameters dynamically to accommodate deeper networks. Lipsformer (Qi et al., 2023a) introduce a Lipschitz continuity constraint to ensure stability in transformer networks, addressing the issue of exploding gradients. Large et al. (2024) introduces a modular norm strategy for scalable optimization. The modular norm normalizes the weights and their updates in the forward and the backward individually. They prove that the gradient of the network is Lipschitz-continuous in the modular norm with the Lipschitz constant that admits a simple recursive formula. The modular norm introduces a new possible direction for future deep neural network optimization. However, a problem is that it cannot be directly plugged into current Transformer framework. All components in Transformer needs to be re-adapted. *Our Stable-Transformer is built on our stabilized components, i.e., StableInit, StableNorm and StableAtten. It roots on solid theoretical justification.*

Some other great works also investigate the feature learning or representation learning (Yang, 2019; Yang & Hu, 2021; Yang et al., 2022) and learning stability (Bernstein et al., 2020), we would like to recommend them to the readers although they are not directly related to this paper.

