# OpenReview forum: "Stable-Transformer: Towards a Stable Transformer Training"
_ICLR.cc/2025/Conference — Submitted to ICLR 2025_

### Official Review · Reviewer_6ZEa · 2024-11-02

**Soundness:** 4
**Presentation:** 4
**Contribution:** 4
**Rating:** 6
**Confidence:** 4

**Summary:**

This paper is motivated by the observation that a rigorous mathematical understanding of why training instabilities occur in transformers—and why current stabilization techniques work—is still lacking. To address this, the authors provide a theoretical analysis of transformer initialization, normalization, and attention mechanisms, proposing a set of stabilization methods for each. Experimental results show that these individual stabilizations enhance training stability. Furthermore, by combining these approaches, the authors introduce a new model, the Stable-Transformer, which demonstrates a more stable training process in experiments.

**Strengths:**

1.  The paper is well-organized and clearly articulated. The problem statement, proposed solution, and experimental results are presented in a logical and accessible way. The figures and diagrams effectively illustrate the key concepts and findings.
2.  The paper holds significant value as it thoroughly analyzes the instability issues associated with training transformers. It also introduces a stabilized model, the Stable-Transformer, and demonstrates through experiments that this model achieves a more stable training process.
3. The proposed method is firmly based on robust theoretical analysis. The mathematical formulation and proofs are clearly articulated and well-supported.

**Weaknesses:**

Overall, I'm satisfied with this paper and its core contributions. Other minor concerns/suggestions are listed as follows:

Have the authors considered assessing Stable-Transformer on large multi-modal models such as CLIP or Flamingo? Evaluating these models could showcase the wider applicability of the proposed stabilization techniques beyond just language and vision transformers.

**Questions:**

Considering the encouraging outcomes with GPT and ViT models, what do the authors identify as the key next steps for validating and enhancing Stable-Transformer? Are there specific challenges you foresee in implementing these techniques on larger models or different modalities?

---

> ### Author Response · Authors · 2024-11-21
> **Author response to Reviewer 6ZEa (part 1/2)**
>
> We thank the reviewer very much for highly appreciating the contributions of our work. We observe more and more researchers are focusing on fantastic applications based on Transformer models, but we believe that the research community would also carry the mission of developing mathematical understanding of the Transformer. Unfortunately, it is still not enough. We are very eager to make a little contribution to that. We are really appreciating your recognition of our work. It means a lot to us. We will insist to devote on this direction.
>
> &nbsp;
>
> *__1. Reviewer's comments on minor concerns/suggestions__*
>
> > *Overall, **I'm satisfied with this paper and its core contributions**. Other minor concerns/suggestions are listed as follows:*
> >
> > *Have the authors considered assessing Stable-Transformer on large multi-modal models such as CLIP or Flamingo? Evaluating these models could showcase the wider applicability of the proposed stabilization techniques beyond just language and vision transformers.*
>
> __Our Response__: Thank you very much for your comments. We will consider multi-modality applications, such CLIP, LLaVa, and Stable-Diffusion. In our previous submission, the experiments have cost us many computing resources. Running a StableGPT-small and nanoGPT both cost us around one week on 8 A800 GPUs. StableGPT-medium, StableViT-large, StableViT-huge, and many other ablation studies cost us more resources. It should be noted that training CLIP from scratch usually takes around two weeks on  64 A800 GPUs even with ViT-B-32 models. This is an embarrassing situation for us, but we will try that in the future.
>
> Instead, we have conducted experiments on larger and deeper models, including StableViT-g (1B parameters), StableGPT-large (0.77B) and StableViT-200 (which has 200 layers and 1.44B parameters). Experimental results have further verified that the proposed Stable-Transformer has better stability and effectiveness.
>
> &nbsp;

---

> > ### Comment · Reviewer_6ZEa · 2024-11-26
> >
> > I would like to keep my original score; however, I suggest that the authors conduct experiments on multimodal models to further assess the effectiveness of the proposed methods.

---

> > > ### Author Response · Authors · 2024-11-27
> > > **Author response to Reviewer 6ZEa**
> > >
> > > We sincerely appreciate your high recognition of our paper as an excellent work. Thanks for your valuable suggestions about experiments on multi-modal models, e.g., CLIP or Flamingo.
> > > According to [1], to train CLIP with ViT-L/14 for 32 epochs using data set LAION-400M with batch size 32K, it will take one month on 256 A800s. During the rebuttal period, we can neither get the data ready nor find enough computing resources. We will put the task of conducting multi-modality experiments as our future work and will try the best of our team to fulfill it. Thank you again for your support.
> > >
> > > [1] Xu, Hu, Saining Xie, Xiaoqing Ellen Tan, Po-Yao Huang, Russell Howes, Vasu Sharma, Shang-Wen Li, Gargi Ghosh, Luke Zettlemoyer, and Christoph Feichtenhofer. "Demystifying clip data." arXiv preprint arXiv:2309.16671 (2023).

---

> ### Author Response · Authors · 2024-11-21
> **Author response to Reviewer 6ZEa (part 2/2)**
>
> *__2. Reviewer's questions on the key next steps__*
>
> > *Considering the encouraging outcomes with GPT and ViT models, what do the authors identify as the key next steps for validating and enhancing Stable-Transformer? Are there specific challenges you foresee in implementing these techniques on larger models or different modalities?*
>
> __Our Response__: Frankly, the next key step to validate our proposed Stable-Transformer is to conduct more experiments on some larger language models (e.g., 7B), and some multi-modality large model, e.g., CLIP or Flamingo, as the reviewer pointed out. Moreover, we are also interested to validate our proposed Stable-Transformer to the senario when the sequence length is very long (e.g., 10k) or the data are corrupted by adversary noises, to figure it out that whether there are other instability risks in training large language model or multi-modal large language model.
>
> Embarrassingly, the biggest challenge we currently met is the shortage in computing resources. It needs hundreds of GPUs to train tens of weeks on billions of text-image pairs, such as LAION.
>
> &nbsp;
>
> Finallym we sincerely thank reviewer 6ZEa for giving us valuable suggestions, which have greatly helped improve our paper. We hope that our responses would address your concerns above. We are very happen to response if there are any other concerns or suggestions.
>
> &nbsp;

---

> > ### Author Response · Authors · 2024-11-25
> > **Sincerely invite you to check our revisions**
> >
> > Thank the reviewer for the time and effort in reviewing our paper. We have largely strengthened the paper according to your comments. We hope you will be satisfied with our revisions. We sincerely hope you will reconsider your rating based on our revised paper. Thank you for your time.
> >
> > Best regards
> >
> > Authors

---

### Official Review · Reviewer_yoj2 · 2024-11-02

**Soundness:** 3
**Presentation:** 3
**Contribution:** 2
**Rating:** 6
**Confidence:** 4

**Summary:**

The paper introduces several techniques to stabilize transformer training: StableInit, StableNorm, and StableAtten, which replace Xavier initialization, LayerNorm, and Attention, respectively. For each component, the authors first identify limitations in previous methods and key factors contributing to stability, then propose the solutions. They provide theoretical analyses and experimental validation of each technique on GPT2-S and ViT backbones, demonstrating improved training stability over baselines. Additionally, the supplementary materials show that these stabilization techniques enable training with larger learning rates.

**Strengths:**

1. The authors identify instability issues in each vanilla module when training large models and propose insights to address them.

2. Experiments on GPT and ViT verify that the proposed approaches enhance training stability, with each experiment and theoretical analysis supporting the stability claims.

3. The authors demonstrate that these methods enable the network to tolerate larger learning rates.

4. The paper is well-written, with a clear structure that makes it easy to follow.

**Weaknesses:**

1. While each method contributes to improved stability, the gains from each proposed module are incremental. It would strengthen the argument to demonstrate that alternative methods, such as enforcing Lipschitz continuity in normalization [1], cannot replace each proposed module (e.g., showing that StableNorm outperforms other normalization techniques). Comparing the proposed methods with alternative stability techniques, rather than just naive baselines, would be more compelling. How does StableNorm compare to other recent normalization techniques like DeepNorm? And how does StableAtten compare to L2 Self-Attention [2]?

2. Although these methods make training more stable and tolerant of larger learning rates, the authors do not present other potential benefits of stable training. Emphasizing the importance of stability improvements would strengthen the argument. Does improved stability benefit generalization, lead to faster convergence, or improve performance on out-of-distribution data?

3. It would be valuable to test stability across an increasing number of layers and demonstrate whether improved stability benefits downstream tasks or enhances robustness to sample or label noise. Can the proposed techniques enable training with more layers, such as 200 or 1000 layers? Does improved stability benefit fine-tuning ViT for image classification on smaller datasets like CIFAR-10/100, or make it more robust to distribution shifts, such as on CIFAR-10-C and CIFAR-100-C?



[1] Gouk et al., "Regularisation of neural networks by enforcing Lipschitz continuity," Machine Learning, 2021.

[2] Kim et al., The Lipschitz Constant of Self-Attention, ICML 2021.

**Questions:**

Please see the weaknesses.

---

> ### Author Response · Authors · 2024-11-21
> **Author responses to Reviewer yoj2 (part 1/2)**
>
> We would like to express our sincere thanks to the reviewer for the insightful comments to the paper's strengths. We highly appreciate the reviewer for recognizing the contributions of our paper. We made a point-by-point responses to the comments, added experiments with  **StableViT-200, which is a network of 200 layers**, and we also conducted experiments with **StableViT-g (1B parameters)** and **StableGPT-large (0.77B)**.
>
> &nbsp;
>
> *__1. Reviewer's comment on normalization__*
>
> > *While each method contributes to improved stability, the gains from each proposed module are incremental. It would strengthen the argument to demonstrate that alternative methods, such as enforcing Lipschitz continuity in normalization [1], cannot replace each proposed module (e.g., showing that StableNorm outperforms other normalization techniques). Comparing the proposed methods with alternative stability techniques, rather than just naive baselines, would be more compelling. How does StableNorm compare to other recent normalization techniques like DeepNorm? And how does StableAtten compare to L2 Self-Attention [2]?*
>
>
> __Our response__: The mentioned prior works address the training instability from different aspects, e.g., DeepNorm only takes into account the aspect of the instability issue from the normalization layer, but ignoring the instability risks from the self-attention module; L2 self-attention only takes into account the aspect of the instability issue from the self-attention module, but ignoring instability risks from initialization and normalization. ** However, our work is to provide a complete set of key techniques---StableInit, StableNorm, StableAtten---to address the instability issues from three different aspects in training a Transformer.** By integrating the three components together, we can alleviate all the training instability risks from the three aspects (i.e., the initialization, normalization, and attention).
>
> Our StableNorm is built on RMSNorm. When $\alpha=0.5$, StableNorm degrades into RMSNorm. For larger models, we can choose a smaller $\alpha<0.5$ to make the network more stable.  In previous submission, we compare StableNorm with RMSNorm and LayerNorm. Why we choose a RMSNorm as our baseline is because RMSNorm is the currently most used normalization in large language model (such as PaLM, LLaMA1, LLaMA2 and LLaMA3).
>
> References [1, 2] are two excellent works that enfores Lipschitz continuity in different modules. There is an interesting work, termed LipsFormer, which is highly inspired by the two papers [1, 2]. We also believe that the Lipschitz continuity of the network is a very important condition for a stable training. Actually the principle behind StableNorm can also be explained by Lipschitz constant. The Jacobian matrix of StableNorm is defined as:
>
> $$
> \frac{\partial \operatorname{StableNorm}(\boldsymbol{x})}{\partial \boldsymbol{x}} = \frac{ {d^{\alpha}}} {{ \sqrt{{\lVert \boldsymbol{x} \rVert}_2^2 + \epsilon}}} \left(\boldsymbol{I}-\frac{\boldsymbol{x} \boldsymbol{x}^{\top}}{\lVert\boldsymbol{x}\rVert_2^2 + \epsilon }\right) \operatorname{diag}(\boldsymbol{\gamma}).
> $$
>
> and the Jacobian matrix of RMSNorm is defined as:
>
> $$
> \frac{\partial \operatorname{RMSNorm}(\boldsymbol{x})}{\partial \boldsymbol{x}} = \frac{ {d^{0.5}}} {{ \sqrt{{\lVert \boldsymbol{x} \rVert}_2^2 + \epsilon}}} \left(\boldsymbol{I}-\frac{\boldsymbol{x} \boldsymbol{x}^{\top}}{\lVert \boldsymbol{x}\rVert_2^2 + \epsilon }\right) \operatorname{diag}(\boldsymbol{\gamma}).
> $$
>
> by choosing a smaller $\alpha <0.5$, the Lipschitz constant of StableNorm is less than that of RMSNorm. For example, if $d=1024$, when we choose $\alpha=0.475$, the Lipschitz constant of StableNorm is only around 84% ($\frac{1024^{0.475}}{1024^{0.5}}$) of that of RMSNorm. This also explains why StableNorm has a better stability than RMSNorm. DeepNorm is an excellent work, but we do not include it for a comparison because it is not easy for us to include in into nanoGPT and Timm framework since that it needs specific parameter choices, i.e., $\alpha$ and $\beta$, for the initialization. It seems that for different networks, the parameters $\alpha$ and $\beta$ are different. We do not know how to choose a proper $\alpha$ and $\beta$ for nanoGPT and ViT in timm.
>
> L2 self-attention is the first work that studies Lipschitz constant of self-attention and inspires a lot of later works. According to our experiments, we find that StableAtten achieves a better performance than L2 self-attention. We think the reason behind the results is that in StableAtten, we will normalize the query $Q$ and the key $K$, it will largely constraint the Lipschitz constant to a controllable range according to the gradient computation in the paper. But we observe that L2 self-attention still has stability issue especially when $W_q \neq W_k$. This may be due to L2 self-attention does not fully resolve the influence brought by normalization.

---

> ### Author Response · Authors · 2024-11-21
> **Author responses to Reviewer yoj2 (part 2/2)**
>
> ***2. Reviewer's comment on other potential benefits***
>
> > *Although these methods make training more stable and tolerant of larger learning rates, the authors do not present other potential benefits of stable training. Emphasizing the importance of stability improvements would strengthen the argument. Does improved stability benefit generalization, lead to faster convergence, or improve performance on out-of-distribution data?*
>
> __Our response__: Training instability is an important issue in large model, as mentioned in PaLM [1]: "For the largest model, we observed **spikes in the loss roughly 20 times during training, despite the fact that gradient clipping was enabled.**
>
> From our experiments on StableGPT and StableViT, we can find that they both outperform their counterpart, i.e., GPT and ViT. We have observed that enabling larger learning rates can lead to better performance. Our StableGPT-small with a larger learning rate 1.2e-3 can have a better validation loss, i.e., 2.819 vs 2.827,  compared to that with a learning rate 6e-4. Since that we did not largely modify the network structure, it is hard to expect faster convergence. But we did find out that choosing the activation function from GLU to SwiGLU can largely improve the performance. However, we did not include the result in our paper due to the fact that such a modfication suffers from the instability risk. We believe that our StableNorm and StableAtten will benifit the LLM researchers for training large language models or multi-modality models.
>
> We really hope the reviewer would be satisfied our point-to-point responses. Thank you!
>
> [1] Chowdhery, Aakanksha, et al. "Palm: Scaling language modeling with pathways." *Journal of Machine Learning Research* 24.240 (2023): 1-113.
>
> &nbsp;
>
> ***3. Reviewer's comment on deeper layers***
>
> > *It would be valuable to test stability across an increasing number of layers and demonstrate whether improved stability benefits downstream tasks or enhances robustness to sample or label noise. Can the proposed techniques enable training with more layers, such as 200 or 1000 layers? Does improved stability benefit fine-tuning ViT for image classification on smaller datasets like CIFAR-10/100, or make it more robust to distribution shifts, such as on CIFAR-10-C and CIFAR-100-C?*
>
> __Our response__: thank you very much for your suggestion. We have tested the StableGPT on larger version. StableGPT-large contains 36 layers instead of 12 layers in StableGPT-small. It exhibits still very stable training process. Meanwhile, we still observe d that it can enable larger learning rate. Moreover, we have tested StableViT-g that has 40 layers. StableViT-g can stably converge even without learning rate warmup, but ViT-g without learning rate warmup will crash after a certain number of training steps.  Furthermore, we have tested StableViT-200 that 200 layers. In this case, the loss of the baseline ViT spikes around the 56th epoch, but our **StableViT-200 can stably converge**. We have included the newly added experiments in the revised paper.
>
> &nbsp;
>
> We sincerely thank the reviewer for your valuable suggestions, which have greatly helped strengthen our paper.
>
> Hope our clarifications and the new experimental results could address your concerns.
>
> &nbsp;

---

> ### Author Response · Authors · 2024-11-25
> **Sincerely invite you to check our revisions**
>
> We would like to thank the reviewer for the time and effort in reviewing our paper. We have addressed your comments and concerns, especially about more comparisons and deeper networks. We hope you will be satisfied with our revisions. We sincerely hope you will reconsider your rating based on our revised paper. Thank you for your time.
>
> Best regards,
> Authors

---

> ### Comment · Reviewer_yoj2 · 2024-11-26
> **Thanks to the Authors' Response**
>
> Thank you for addressing my concerns. While my questions regarding comparisons with other normalization techniques and training with more layers have been resolved, the advantages of StableAttn over other Lipschitz attention methods, as well as the additional benefits of stable training, remain unaddressed. Could the authors provide experimental results comparing StableAttn with other Lipschitz-enforcing attention methods and demonstrate whether improving training stability helps fine-tuning or distribution shifts in downstream tasks?

---

> > ### Author Response · Authors · 2024-11-26
> > **Thank reviewer yoj2 for your kind reply.**
> >
> > Thank you very much for your kind reply. We are happy to know that a part of your concerns have been resolved. Regarding the comparisons to other Lipschitz-enforcing attention methods, and about whether improving training stability helps fine-tuning or distribution shifts in downstream tasks, we have some comparison results already now, and some experiments still running. We expect to provide a set of complete results tomorrow. We will give you a reply tomorrow and provide an updated paper accordingly. Thank you again for your feedback!

---

> > ### Author Response · Authors · 2024-11-27
> > **Author responses to Reviewer yoj2**
> >
> > Thank you for your comments. To further address your concerns, we have conducted experiments from the following two aspects,
> >
> > - we conducted experiments to compare StableAtten with $L_2$ self-attention under two different settings (i.e., $\boldsymbol{W}_q = \boldsymbol{W}_k$ and $\boldsymbol{W}_q \neq \boldsymbol{W}_k$) and have added the experimental results **in Appendix N**. Please take a look at Figure 14.
> >
> > - we conducted experiments to compare our StableViT and ViT on the clean CIFAR-100 and the corrupted CIFAR-100-C and have added the experimental results **in Appendix O.** Please check Table 4.
> >
> > We have updated our paper, and highlighted the revised contents in red. We sincerely thank the reviewer again for the valuable suggestions and insightful comments, which have greatly strengthened the quality of our work.
> >
> > We hope you will be satisfied with our responses and our revisions, and **we really hope you will reconsider your rating**. we are happy for any further discussions.

---

> > > ### Comment · Reviewer_yoj2 · 2024-11-28
> > >
> > > Thank you for the author's response. After reviewing the revised paper, I believe most of my concerns have been addressed, and I will raise my score to 6.

---

### Official Review · Reviewer_xFs7 · 2024-11-03

**Soundness:** 2
**Presentation:** 2
**Contribution:** 3
**Rating:** 5
**Confidence:** 2

**Summary:**

This paper proposes theoretical and experimental approaches to improve the training stability of Transformer models. To address the instability issues arising from the expansion of parameters in large-scale Transformers, the authors introduce new initialization (StableInit), normalization (StableNorm), and attention mechanisms (StableAtten). These three techniques are combined to form a "Stable-Transformer," which the authors demonstrate experimentally to improve training stability and performance.

**Strengths:**

Theoretical Contribution: The paper provides a theoretical analysis of the training instability in Transformers and proposes improvements to initialization, normalization, and attention mechanisms to address this instability. This is a significant contribution to the understanding and enhancement of Transformer model training.

Novel Stabilization Techniques: The authors propose StableInit, StableNorm, and StableAtten to stabilize the initialization, normalization, and attention mechanisms, respectively. By combining these, the authors offer a comprehensive solution to stabilize the entire Transformer architecture.

**Weaknesses:**

Lack of Scalability Validation for Large Models: While the paper provides a theoretical foundation and experimental validation of the stabilization techniques, it lacks empirical testing for scalability to larger models, such as massive language models.
Implementation Complexity of StableNorm and StableAtten: The proposed techniques, StableNorm and StableAtten, require additional hyperparameter tuning, which may increase implementation complexity. This complexity can limit the practical applicability of these methods in real-world scenarios.
Limited Experimental Scope: The experiments are limited to GPT and ViT, with no validation across a broader range of Transformer architectures or real-world applications. This raises questions about the generalizability of the proposed stabilization techniques.

**Questions:**

see weakness

---

> ### Author Response · Authors · 2024-11-21
> **Author responses to Reviewer xFs7 (part 1/2)**
>
> We thank the reviewer for appreciating our work as "**a significant contribution to the understanding and enhancement of Transformer model training**", and offering "**a comprehensive solution to stabilize the entire Transformer architecture**".
>
> To address the concerns of the reviewer, we have added more experimental results, especially on __large and deep models__ (including StableViT-g (1B parameters), StableGPT-large (0.77B) and StableViT-200 which has 200 layers and 1.44B parameters). Experimental results further verified that the proposed Stable-Transformer has better stability and effectiveness. We have added these new experiments in the revised version of our paper. We hope the reviewer would be satisfied with our revision and we would be very glad to address any further concerns or comments.
>
> &nbsp;
>
> ***1. Reviewer's main concern on experiments on larger models***
>
> > _Lack of **Scalability Validation for Large Models**: While the paper provides a theoretical foundation and experimental validation of the stabilization techniques, it lacks empirical testing for scalability to larger models, such as massive language models._
>
> **Our Response:**  thank you very much for your comments. To address your concern, we have conducted experiments on more large-scale models in the revised version of our paper according to your suggestions. We would like to clarify our added experiments.
>
> __Larger StableViT (1B parameters)__
>
> - In the revised paper, we further verify a larger version of StableViT, we term it as StableViT-g and compare it with ViT-g. Both StableViT-g and ViT-g have around 1.0 billion of parameters. The parameter scale of the StableViT-g is __116 times__ of ViT-base (86M), and __113 times__ of Swin-Transformer-Base (88M).
> - From Figure 9 and Figure 10 in the revised paper, we can observe that our StableViT-g consistently outperforms ViT-g. Meanwhile, we note that the training of StableViT-g can stably converge even without learning rate warmup; whereas without the learning rate warmup, ViT-g will crash after a certain number of training steps. Even with the learning rate warmup, the loss of ViT-g will have a spike. This has further verified the stability of our proposed method on larger models.
>
> __Larger StableGPT (0.77B parameters)__
>
> - We have also conduct experiments on larger StableGPT (0.77B), termed as StableGPT-large. We would like to point out in our previous submission, it takes around one week to run StableGPT-small (124M=0.124B) on 8 A800 GPUs. We would also like to point out that training a model with 8 A100 GPUs is a luxury thing even for a top university. We run StableGPT-large on 16 A800 GPUs and compare it with its counterpart, GPT2-large. We believe that the parameter scale is large enough to verify the effectiveness of a GPT algorithm.
>
> - In Figure 15 of the revised paper, we can see that StableGPT-large consistently outperforms GPT-large. StableGPT-large can stably converge and achieve a better performance. It verified the stability of our StableGPT.
>
> **Deeper and Larger StableViT (1.44B parameters)**
>
> - We have further verified a deeper and larger version of StableViT, which has 200 layers and thus we term it as StableViT-200. We compare it with ViT-200. We run StableViT-200 on 16 A800 GPUs and use gradient accumulation to keep the global batch size as 1024. It takes around 3 days for ViT-200 to run 60 epochs.
>
> - In Figure 12 and Figure 13 of the revised paper, we can see that, with learning rate warmup, ViT-200 has a smoothing loss curve in the early stage, while the loss has a spike around the 56th epoch. But StableViT-200, without using learning rate warmup, can converge stably. It fully verified the stability of StableViT in a very deep Transformer.
>
> We kindly invite the reviewer to check the newly added **Figures 9, 10, 11, 12, and 13** in __Appendix J, K, L__ in the revised paper.
>
> &nbsp;

---

> ### Author Response · Authors · 2024-11-21
> **Author responses to Reviewer xFs7 (part 2/2)**
>
> ***2. Reviewer's comment on computational complexity***
>
> > _Implementation Complexity of StableNorm and StableAtten: The proposed techniques, StableNorm and StableAtten, require additional hyperparameter tuning, which may increase implementation complexity. This complexity can limit the practical applicability of these methods in real-world scenarios. Limited Experimental Scope: The experiments are limited to GPT and ViT, with no validation across a broader range of Transformer architectures or real-world applications. This raises questions about the generalizability of the proposed stabilization techniques._
>
> **Our response**: Thank you very much for your comments. We would like to clarify the following points.
>
> 1. The computation complexity of StableNorm is the same as RMSNorm. The definition of StableNorm is:
>
>    $\operatorname{StableNorm}(\boldsymbol{x})  = \boldsymbol{\gamma} \odot \frac{ d^{\alpha} \boldsymbol{x}}{\sqrt{{| \boldsymbol{x} |}_2^2 + \epsilon}}$
>
>    when $\alpha =0.5$, the StableNorm degrades to RMSNorm. In very large model, since that the hidden dimension $d$ is very large, using a smaller $\alpha < 0.5$ (such as 0.475 or 0.45) will make the gradients smaller so the probability to triger the gradients exploding in the back-propagation process would be small. On the other hand, the forward and backward computational complexities between StableNorm and RMSNorm are exactly same.
>
> 2. In StableAtten, we will normalize the query $Q$ and the key $K$ before computing the softmax, which has the same computational cost as QK-Norm. Even compared with a vanilla attention, it will only bring in less than 10% overall compuation cost.  As a return, our StableAtten will lead to very smoothing training process.
>
> 3. StableInit has the same computational cost as the Xavier initialization. It is only applied once before the model training.
>
> 4. In our model, there are only a single parameter $\alpha$ in all StableNorm, StableInit and StableAtten, which is needed to be choosen and all others are fixed. For larger model, a smaller $\alpha$ can be used to achieve better stability.
> We believe that it brings better maneuverability and stability to LLM and large vision model, and thus the researchers in LLM community will benifit from it.
>
> &nbsp;
>
> We really hope the reviewer would be satisfied with our revisions. We are very glad to discuss further if there are any question or concerns.

---

> ### Author Response · Authors · 2024-11-25
> **Sincerely invite you to check our revisions**
>
> We would like to thank the reviewer for the time and effort in reviewing our paper. We have addressed your comments and concerns in the revised paper. We hope you might find our responses satisfactory. We sincerely hope you will reconsider your rating based on our revised paper. Thank you for your time.
>
> Best regards,
>
> Authors

---

> > ### Comment · Reviewer_xFs7 · 2024-11-26
> > **Thank you for Response**
> >
> > Thank you for helping me resolve my doubts. I believe there will be an improvement in the score.

---

> > > ### Author Response · Authors · 2024-11-27
> > > **Author responses to Reviewer xFs7**
> > >
> > > We thank the reviewer for the valuable time and the effort in reviewing our paper.
> > >
> > > In the rebuttal, we have further conducted experiments to compare StableAtten with $L_2$ self-attention under two different settings ($\boldsymbol{W}_q = \boldsymbol{W}_k$ and $\boldsymbol{W}_q \neq \boldsymbol{W}_k$) **in Appendix N**, and also compared our StableViT and ViT on the clean CIFAR-100 and the corrupted CIFAR-100-C **in Appendix O.**
> > >
> > > We have updated our paper, and highlighted the revised contents in red. We hope you might find our responses and revisions satisfactory. **We sincerely hope you will reconsider your rating based on our clarification in responses and the revised paper**. Thank you for your time!

---

> > > ### Author Response · Authors · 2024-11-29
> > > **Thank you very much for your time and effort**
> > >
> > > Dear Reviewer xFs7,
> > >
> > > Thank you very much for your time and effort. Since you have acknowledged the contributions of our work and confirmed that your intially raised concerns have been comprehensively addressed. We would greatly appreciate if you could reconsider your score for our submission. Your support means a lot to us.
> > >
> > > Best regards
> > >
> > > Authors

---

> > > ### Author Response · Authors · 2024-12-02
> > > **A kind remind**
> > >
> > > Dear Reviewer xFs7,
> > >
> > > Thank you very much for your time and effort. Since the discussion phase is approaching to close, we are very much looking forward to hearing from you. We sincerely hope you will reconsider your rating based on our clarification and the revised paper.
> > >
> > > Best regards
> > >
> > > Authors

---

### Author Response · Authors · 2024-11-21
**Summary of our revisions**

We would like to thank all reviewers for their valuable time and great effort in reviewing our paper!

To address reviewers' comments and concerns, we have made the following changes:

- We added experiments on a larger vision Transformer, i.e., StableViT-giant that has 1B parameters, in **Appendix J**. (xFs7)

- We added experiments on a larger language Transformer, i.e., StableGPT-large that has 774M (0.774B) parameters, in **Appendix K**. (xFs7, 6ZEa)

- We added experiments on a 200-layer vision Transformer, i.e., StableViT-200 that has 1.44B  parameters, in **Appendix L**. (yoj2, xFs7)

- Based on the newly added experiments, we further clarify our practical contribution in the abstract and introduction section.

The revised content is highlighted in purple in the revised paper. We kindly invite the reviewer to check the newly added experiments.

We sincerely thank all reviewers again for their valuable suggestions and insightful comments, which have greatly helped strengthen our paper. For any further questions or concerns, we would be happy to discuss them further.

---

### Public Comment · ~Yuzhu_Wang1 · 2024-11-27
**Some  confusion about Fig.1**

Dear authors:

I read the ICLR submission with great interest. I think it provides a good contribution on training Transformers. But I have some confusion about Fig.1, and I look forward to the authors providing explanations.

(1) Why does the accuracy of baseline ViT-L-150-epoch suddenly increase at about 60 epochs?

(2) Why does the accuracy of StableViT suddenly exceed the baseline at the end of training, for example, at 250 epochs? Training 150 epochs has similar phenomena.

(3) Note that the classic Masked AutoEncoder provides training recipes and reports 82.6% accuracy on ImageNet for ViT-L trained from scratch. The result is higher than the baseline (81.3%) and StableViT-L (82.4%). Importantly, the training process ViT-L-300-epoch (green curve of Fig. 1)  showed no instability.

---

> ### Author Response · Authors · 2024-11-27
> **Thank you for your interests and recognizing of our work.**
>
> Thank you very much for your interest and your careful reading of our paper, and we would like to express our sincere thanks for recognizing our contributions. In the following, we will give the point-by-point responses to your questions.
>
>
>
> > (1) Why does the accuracy of baseline ViT-L-150-epoch suddenly increase at about 60 epochs?
>
> As pointed out in our paper (in Lines 1046-1047), we follow the implementation of Adan [1]. We quote our description in our submitted paper here: "Note that in the original ViT, we use 60 epochs’ learning rate warmup, but in our StableViT, we do not use warmup."
>
> The performance of ViT-L suddenly increases because the learning rate warmup stage just ends. The sudden change of learning rate from (learning rate) increasing to (learning rate) decreasing is the reason caused the sudden change.
>
>
>
> > (2) Why does the accuracy of StableViT suddenly exceed the baseline at the end of training, for example, at 250 epochs? Training 150 epochs has similar phenomena.
>
> Thank you for your question. To be honest, the training process of Transformer is a complex dynamic process. It is hard for us to give you a precise answer. We believe that the phenomena is due to the difference in stability of the original Transformer and our Stable-Transformer. Since that the loss landscape of the original Transformer is steeper; whereas the loss landscape of our Stable-Transformer is smoother. Therefore, the original Transformer may go into a local optimum quickly and then tends to saturate, but our Stable-Transformer may converge more smoothly. As a sequence, in the late stage of the training process, our Stable-Transformer can increase further higher.
>
>
>
> > (3) Note that the classic Masked AutoEncoder provides training recipes and reports 82.6% accuracy on ImageNet for ViT-L trained from scratch. The result is higher than the baseline (81.3%) and StableViT-L (82.4%). Importantly, the training process ViT-L-300-epoch (green curve of Fig. 1) showed no instability.
>
> Thank you for your question. We would like to quote the description in the paper of Masked AutoEncoder [2]: "The accuracy is 82.6% for ViT-L (81.5% w/o EMA)", in page 12 and just below Table 11 in the Appendix section. In our paper, we have pointed out that we do not use EMA for the experiments reported in Table 3 of our paper. Thus, the accuracy of VIT-L without using EMA in our submission is 81.3%, and the result of ViT-L without using EMA in [2] is 81.5%. The slight difference should be due to some minor discrepancy in implementation details, e.g. weight decay and batch size. In summary, the overall results reported in our paper and that in [2] are consistent.
>
> The result of ViT-L-300 in Figure 1 is with a learning rate warmup of 60 epochs. You can refer to Figures 2, 3 and 4 for unstable cases without using learning rate warmup.
>
>
>
> Thank you again for your questions. We hope our replies will resolve your doubts.
>
>
>
> [1] Xie, X., Zhou, P., Li, H., Lin, Z., & Yan, S. (2024). Adan: Adaptive nesterov momentum algorithm for faster optimizing deep models. *IEEE Transactions on Pattern Analysis and Machine Intelligence*.
>
> [2] He, Kaiming, et al. "Masked autoencoders are scalable vision learners." *Proceedings of the IEEE/CVF conference on computer vision and pattern recognition*. 2022.

---

> > ### Public Comment · ~Yuzhu_Wang1 · 2024-11-29
> > **Thanks**
> >
> > Thanks to the author for answering my doubts.
> >
> > Good luck to you.

---

> > > ### Author Response · Authors · 2024-11-29
> > > **Author reply**
> > >
> > > Thank you for your interests.
> > >
> > > Best regards,
> > >
> > > Authors

---

### Meta-Review · Area_Chair_2RCP · 2024-12-22

**Metareview:**

This paper proposes a set of techniques for stable Transformer training: StableInit, StableNorm and StableAtten. StableInit initializes parameters differently from the Xavier initialization, motived by a Lipschitz constraint; StableNorm generalizes RMSNorm by introducing an additional hyper parameter alpha, and StableAtten builds on top of QKNorm and StableNorm which normalizes Q K and scales the attention logits differently. Experiments are ran on GPT2 alike LM training and ViTs. This paper receives borderline ratings and it is indeed a borderline paper after the AC's assessment.

First and foremost, stabelizing Transformer training is a very important task, but also one that has a high bar for showing meaningful progress, and there are plenty of techniques that have been presented in the literature. Therefore, we need to carefully evaluate if this paper's contribution convincingly justifies the claim of a "Stable Transformer". Now if we look at the techniques, they are quite simple -- you can think of them as rescaling either the initial scale of parameters (StableInit), or the outputs activations at certain locations (StableNorm and StableAtten). In particular, in doing so, it introduces another hyper parameter alpha, which is shown to be important to performance -- and I find the reliance on this hyper parameter unsatisfying. Of course, a method should not be criticized for its simplicity (because it can be a plus if the results are outstandingly good). Now if we look at evaluations, there are much left to be desired. Essentially, there is a lack of comprehensive comparisons to other stable training techniques, also the scope of the evaluation seems to be quite limited (there are two tasks, but again the claim of stable Transformer is much broader than what these two tasks reflect, eg what about encoder decoder tasks or other modalities). I find it pretty hard to draw a clear conclusion that the proposed techniques are necesarily better than what we already know.

Based on these considerations, I don't think this work is ready for publication. I suggest either redefining the scope and claims of this work -- either to focus on a more specific aspect of Transformer training than claiming a general stable Transformer, and/or including more comprehensive comparisons to the literature.

**Additional Comments On Reviewer Discussion:**

The authors did a reasonable good job in answering the clarification questions, but the issues on the significance of evaluations remain.

---

### Decision · Program_Chairs · 2025-01-22

Reject